# The USP46 complex deubiquitylates LRP6 to promote Wnt/β-catenin signaling

Victoria H. Ng [1,2,13], Zachary Spencer[3,13], Leif R. Neitzel[1,4], Anmada Nayak[5], Matthew A. Loberg [6], Chen Shen[5], Sara N. Kassel [1], Heather K. Kroh[6], Zhenyi An[7,8], Christin C. Anthony[1], Jamal M. Bryant[1], Amanda Lawson [1], Lily Goldsmith[1], Hassina Benchabane[3], Amanda G. Hansen[1,9], Jingjing Li [1], Starina D'Souza[1], Andres M. Lebensohn [10], Rajat Rohatgi [11], William A. Weiss [7,8], Vivian L. Weiss [6], Charles Williams[4], Charles C. Hong [4], David J. Robbins [5], Yashi Ahmed [3] ✉ & Ethan Lee [1,2,12] ✉

The relative abundance of Wnt receptors plays a crucial role in controlling Wnt signaling in tissue homeostasis and human disease. While the ubiquitin ligases that ubiquitylate Wnt receptors are well-characterized, the deubiquitylase that reverses these reactions remains unclear. Herein, we identify USP46, UAF1, and WDR20 (USP46 complex) as positive regulators of Wnt signaling in cultured human cells. We find that the USP46 complex is similarly required for Wnt signaling in *Xenopus* and zebrafish embryos. We demonstrate that Wnt signaling promotes the association between the USP46 complex and cell surface Wnt coreceptor, LRP6. Knockdown of USP46 decreases steady-state levels of LRP6 and increases the level of ubiquitylated LRP6. In contrast, overexpression of the USP46 complex blocks ubiquitylation of LRP6 by the ubiquitin ligases RNF43 and ZNFR3. Size exclusion chromatography studies suggest that the size of the USP46 cytoplasmic complex increases upon Wnt stimulation. Finally, we show that USP46 is essential for Wnt-dependent intestinal organoid viability, likely via its role in LRP6 receptor homeostasis. We propose a model in which the USP46 complex increases the steady-state level of cell surface LRP6 and facilitates the assembly of LRP6 into signalosomes via a pruning mechanism that removes sterically hindering ubiquitin chains.

The Wnt/β-catenin (henceforth Wnt) signaling pathway is an evolutionarily conserved pathway that is critical for normal development. Dysregulation of the pathway leads to human diseases such as cancer[1]. In the absence of a Wnt ligand, cytoplasmic β-catenin is maintained at low levels via its assembly into a destruction complex composed of Axin, glycogen synthase kinase 3 (GSK3), casein kinase I α (CKIα), and the tumor suppressor adenomatous polyposis coli (APC). Within this complex, β-catenin is phosphorylated and undergoes ubiquitin-mediated proteasomal degradation[2,3]. In the current model of Wnt activation, binding of Wnt ligands to the Frizzled and LRP5/6 coreceptors induces phosphorylation of LRP5/6 and the formation of

active aggregated receptors or signalosomes. The formation of signalosomes inhibits cytosolic degradation of β-catenin via recruitment of Axin[4]. Accumulated cytoplasmic β-catenin translocates to the nucleus to activate a Wnt transcriptional program[5].

In addition to its key role in controlling cytoplasmic β-catenin levels, ubiquitylation also plays a critical role in maintaining Wnt receptor homeostasis. Ubiquitylation and turnover of Frizzled and LRP5/6 occur via the transmembrane ubiquitin ligases RNF43 and ZNRF3, which promote their turnover[6]. Furthermore, conditional deletion of RNF43/ZNRF3 in mice leads to a marked expansion of intestinal crypts, indicating hyperactivation of the Wnt pathway[7].

Conditional loss of ZNRF3 in the mouse leads to Wnt activation and adrenal hyperplasia[8].

Mutations in RNF43 have been found in 18% of human colorectal adenocarcinomas and endometrial carcinomas[9]. Gene fusions involving R-spondins, which result in their elevated expression, have been found in ~10% of colorectal cancers[10]. Secreted R-spondins (RSPOs) amplify Wnt signaling by inhibiting RNF43/ZNRF3, thereby blocking Wnt receptor ubiquitylation and turnover. Thus, regulation of Wnt receptor levels is crucial for maintaining normal Wnt pathway activity and tissue homeostasis, and their dysregulation has important implications for human disease.

The deubiquitylase(s) that oppose the action of RNF43/ZNRF3 in the Wnt pathway remains unclear. USP8 and USP6 have been shown to oppose the actions of RNF43 and ZNRF3 when expressed in cultured cells[11,12]. However, their activities do not appear to be regulated by Wnt signaling. The USP19 deubiquitylase has been shown to act on LRP6, but its role seems to be primarily involved in regulating its movement from the endoplasmic reticulum[13]. Another deubiquitylase, USP42, has been shown to facilitate RNF43 and ZNRF3 activity by preventing their ubiquitin-mediated turnover[14]. Thus, we sought to identify the deubiquitylase involved in Wnt receptor turnover to gain further insight into how receptor homeostasis is maintained in this pathway.

The USP46 deubiquitylase complex comprises the catalytic USP46 and the WDR40-repeat proteins, WDR20 and UAF1. Previous studies have shown that UAF1 and WDR20 function to stabilize and increase the catalytic activity of USP46[15–19]. Members of the USP46 complex are involved in a range of activities that include increasing the abundance of cell-surface glutamate receptors, modulating AMPA receptor ubiquitination and trafficking, and promoting the proliferation of HPV-transformed cancers by stabilizing the cell cycle regulator, Cdt2[20–22]. Herein, we demonstrate that the USP46 complex is required for Wnt signaling in cultured human cells, Xenopus embryos, zebrafish embryos, and mouse intestinal organoids, indicating evolutionary conservation of function.

We show that in response to Wnt pathway activation, the USP46 complex is recruited to and deubiquitylates cell surface LRP6, blocking its turnover. Together with our size exclusion chromatography and sucrose density centrifugation studies, our data reveal a potential mechanism by which the USP46 complex acts on a newly identified step after LRP6 receptor activation and prior to signalosome formation to control Wnt signaling from the plasma membrane. Our studies suggest a new mechanism by which steady-state levels of LRP6 can be exquisitely regulated and further highlight the importance of receptor homeostasis in the physiological control of Wnt signaling.

## Results

### The USP46 complex is a positive regulator of Wnt signaling
We had previously identified the protein, WDR20, in screens for uncharacterized Wnt components[23,24]. Based on a previous screen for ubiquitin ligase regulators of Wingless (Wg)/Wnt signaling, we initiated a screen for deubiquitylases that regulated the Wnt pathway and identified USP46. USP46 by itself normally exhibits low catalytic activity, but its activity is significantly enhanced when bound to the WD40 repeat proteins, UAF1 and WDR20[25]. Thus, we investigated the effects of the trimeric complex (USP46/UAF1/WDR20) on the Wnt signaling pathway individually and in combination. Using human embryonic kidney (HEK) 293 cells stably transfected with the TOPFlash luciferase-based reporter (HEK293 STF)[26], we found that expressing USP46 complex components individually (Fig. 1A) had minimal effects on Wnt-stimulated reporter activity. Similarly co-expression of two components of the USP46 complex increased Wnt signaling, albeit not significantly (Fig. 1B). In contrast, expression of all three components of the USP46 complex (Tri46) stimulated Wnt signaling even in the absence of exogenous Wnt ligand with dramatic potentiation of signaling and stabilization of β-catenin levels in the presence of Wnt3a (Fig. 1A–C).

We next sought to determine if USP46 is required for Wnt signaling in human cells. Knockdown of USP46 with two independent short-interfering RNA (siRNA) constructs significantly blocked Wnt3a-induced TOPFlash activation (Fig. 1D). Confirming the TOPFlash results, knocking down USP46 also inhibited Wnt3a-induced expression of endogenous Wnt target genes (Supplementary Fig. 1A), providing further evidence that USP46 is required for Wnt-mediated transcription. Knockdown of UAF1 using two different siRNAs similarly reduced Wnt-stimulated activity (Fig. 1E). We noticed co-stabilization of the USP46 complex members when they were co-expressed (Fig. 1A, B). Consistent with this result, loss of UAF1 has been previously shown to decrease USP46 levels[19], and we observed a similar effect when we knocked down UAF1 (Supplementary Fig. 1B). We had difficulty knocking down WDR20 due to the lack of suitable siRNAs that could effectively target its multiple transcripts. We were able to show, however, that knocking out WDR20 by CRISPR-Cas9 editing blocked Wnt3a-induced stabilization of β-catenin six days post-CRISPR treatment (Supplementary Fig. 1C). We obtained similar results when we knocked out USP46 and UAF1 by CRISPR-Cas9 editing. Taken together, our experiments show that the USP46 complex is required for (and acts to potentiate) Wnt signaling.

### The USP46 complex is required for Wnt signaling in Xenopus and zebrafish embryos
We next determined whether the USP46 complex is required for Wnt signaling in a developing organism. Dorsal-anterior structure formation in Xenopus laevis embryos is critically regulated by Wnt signaling[27]. The formation of a secondary axis, which occurs due to ectopic activation of the pathway, represents a robust readout for Wnt signaling in vivo[28,29]. We found that injection of mRNAs encoding individual members of the USP46 complex resulted in a significantly lower number of embryos with duplicated axes compared to embryos co-injected with all three components (Fig. 2A), confirming our findings that all three members of the USP46 complex are required for its full Wnt-potentiating activity in mammalian cells. No duplication was observed in control embryos. These results parallel our observations with Xenopus animal caps (which normally do not exhibit active Wnt signaling), in which injection of mRNAs for all three components significantly induced expression of Xenopus Wnt target genes, Xnr3 and Chordin (Fig. 2B).

To further assess the role of the USP46 complex in regulating Wnt signaling in vivo, we tested the effects of the USP46 complex on Wnt signaling in zebrafish (Danio rerio) embryos. In zebrafish, inhibition of Wnt signaling (both Wnt/β-catenin and noncanonical) leads to cyclocephaly due to the failure of eye precursors to undergo migration[30]. We found that knockdown of each of the USP46 complex components by morpholino oligonucleotide (MO) injection resulted in cyclopic embryos (Fig. 2C) and reduction in endogenous Wnt/β-catenin target gene expression (Fig. 2D), consistent with Wnt/β-catenin pathway inhibition. Significantly, co-injection of MOs with their corresponding mRNAs encoding their respective human gene versions rescued both the cycloptic phenotype and the reduction in Wnt/β-catenin target gene expression (Fig. 2C, D). Our MO studies were confirmed with CRISPRi studies of usp46, uaf1, and wdr20 in zebrafish embryos, and we observed cyclopia upon injections of guide RNAs for all three subunits of the zebrafish USP46 complex (Supplementary Fig. 2). These phenotypic changes were paralleled by decreases in the expression of the Wnt/β-catenin target genes (Supplementary Fig. 3). These results demonstrate that the USP46 complex plays a positive and conserved role in Wnt signaling during early embryonic vertebrate development.

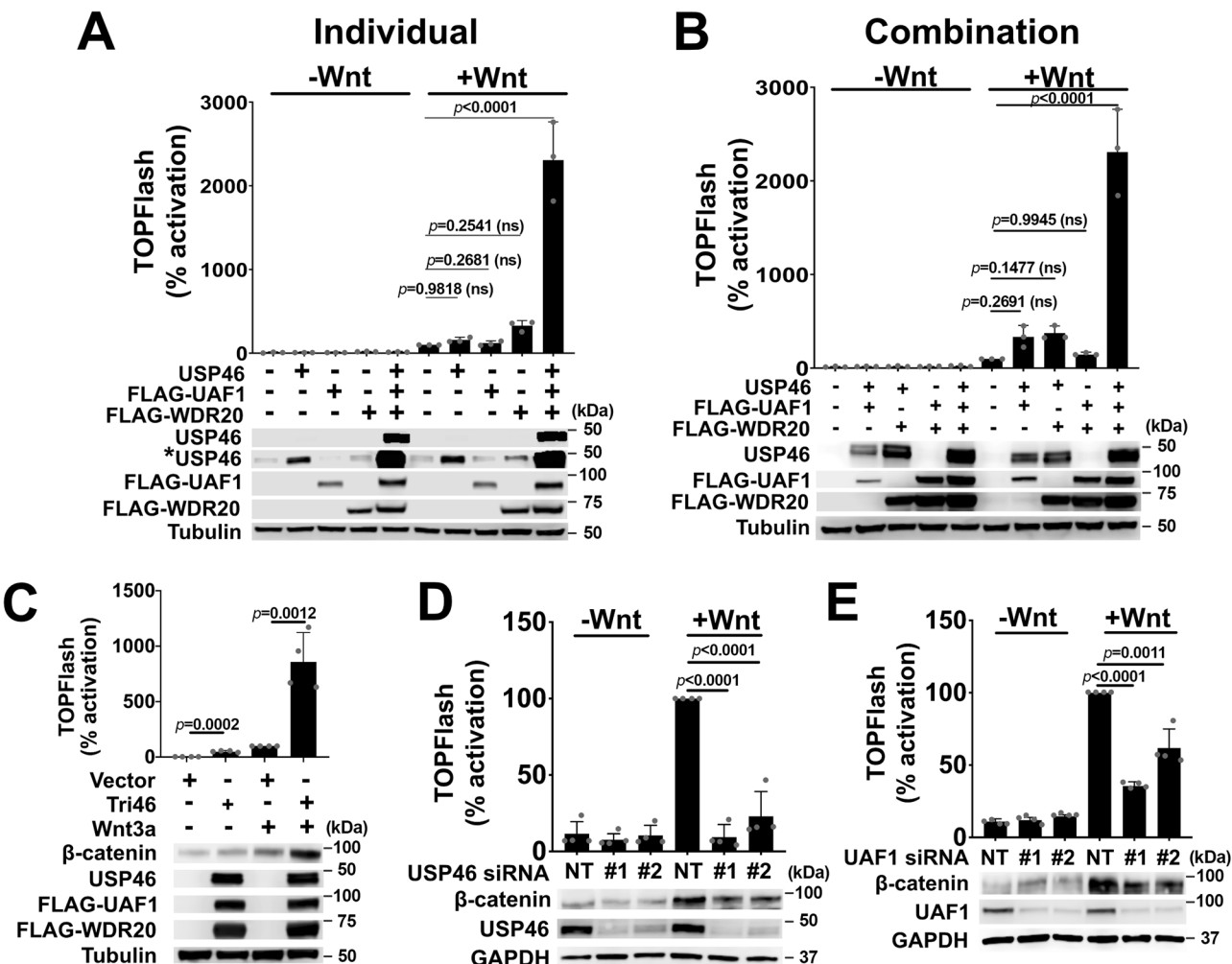

**Fig. 1 | The USP46 complex is a positive regulator of Wnt signaling.**
**A**–**C** Overexpression of the USP46 complex stimulates Wnt activation and stabilizes β-catenin levels. HEK293 STF cells were transfected and treated with recombinant Wnt3a (10 ng/ml) for 24 h as indicated. **A** Individual and **B** pairwise overexpression of USP46, FLAG-tagged WDR20, and FLAG-tagged UAF1 in HEK293 STF cells in the presence of Wnt3a leads to lower Wnt reporter (TOPFlash) activity compared to overexpression of all three members. The asterisk in (**A**) indicates a longer exposure for the USP46 blot. **C** Overexpression of the USP46 complex (Tri46) in HEK293 STF cells potentiates Wnt3a-mediated TOPFlash and promotes β-catenin stabilization. For transfections, the amount of total DNA was kept constant by the addition of vector plasmid. *p* values compare vector-transfected with USP46 complex-

transfected cells. Tubulin is loading control. **D**, **E** Knockdown of USP46 and UAF1 by siRNA inhibits Wnt signaling and decreases β-catenin levels. HEK293 STF cells were transfected with non-targeting control (NT) or two independent (**D**) *USP46* or (**E**) *UAF1* siRNAs and treated with recombinant Wnt3a. Immunoblotting confirmed the knockdown of USP46 and UAF1 proteins. GAPDH is loading control. All graphs show mean ± SD of TOPFlash normalized to vector control in the presence of Wnt. *p* values compare NT treated to *USP46* or *UAF1* siRNA-treated cells. *p* ≥ 0.05 is not significant (ns). Significance was analyzed by one-way ANOVA followed by Tukey's multiple comparisons test. Graphs all show a representative of *n* = 3 biologically independent experiments. All immunoblots are representative of at least three independent experiments. Source data are provided as a Source data file.

## The USP46 complex acts upstream of the destruction complex to increase the steady-state levels of LRP6

Having demonstrated that the USP46 complex is required for Wnt signaling in *Xenopus*, zebrafish, and cultured human cells, we addressed the molecular mechanism of its function in the Wnt pathway. Based on our studies with cultured mammalian cells showing decreased β-catenin levels upon USP46 complex knockdown, it is likely that the USP46 complex acts at or above the level of the β-catenin destruction complex. To test the former possibility, we initially used the GSK3 inhibitor, CHIR99021, to activate Wnt signaling by blocking GSK3-mediated phosphorylation of β-catenin, thereby stabilizing its levels[31–33]. In contrast to our result with Wnt3a treatment, overexpression of the USP46 complex did not potentiate CHIR99021-mediated Wnt activation (Supplementary Fig. 4A). To further test potential roles for USP46 in the destruction complex, we activated the Wnt pathway by knocking down Axin, which is required for assembly of the β-catenin destruction complex. As expected, the knockdown of

Axin increased β-catenin levels (Fig. 3A); this was not affected by knocking down USP46. Tankyrase inhibitors increase steady-state levels of Axin to promote β-catenin degradation and inhibit Wnt signaling[34]. The tankyrase inhibitor, XAV939, blocked the increase in β-catenin levels in cells expressing the USP46 complex in the absence and presence of Wnt3a (Fig. 3B). As a control, XAV939 did not affect the level of an N-terminal truncated form of β-catenin resistant to degradation by the destruction complex (Supplementary Fig. 4B). These experiments show that the USP46 complex likely functions in the Wnt pathway upstream of the destruction complex.

Given that the USP46 complex acts upstream of the destruction complex, we next interrogated the Wnt receptor complex. We were unable to observe any effect on the levels of Frizzled or Dishevelled when we expressed the USP46 complex, although we cannot rule out small changes not detected by our methods. In contrast, we observed increased and decreased levels of LRP6 upon expression and knockdown, respectively, of USP46 in HEK293 cells (Fig. 3C, D). We also found

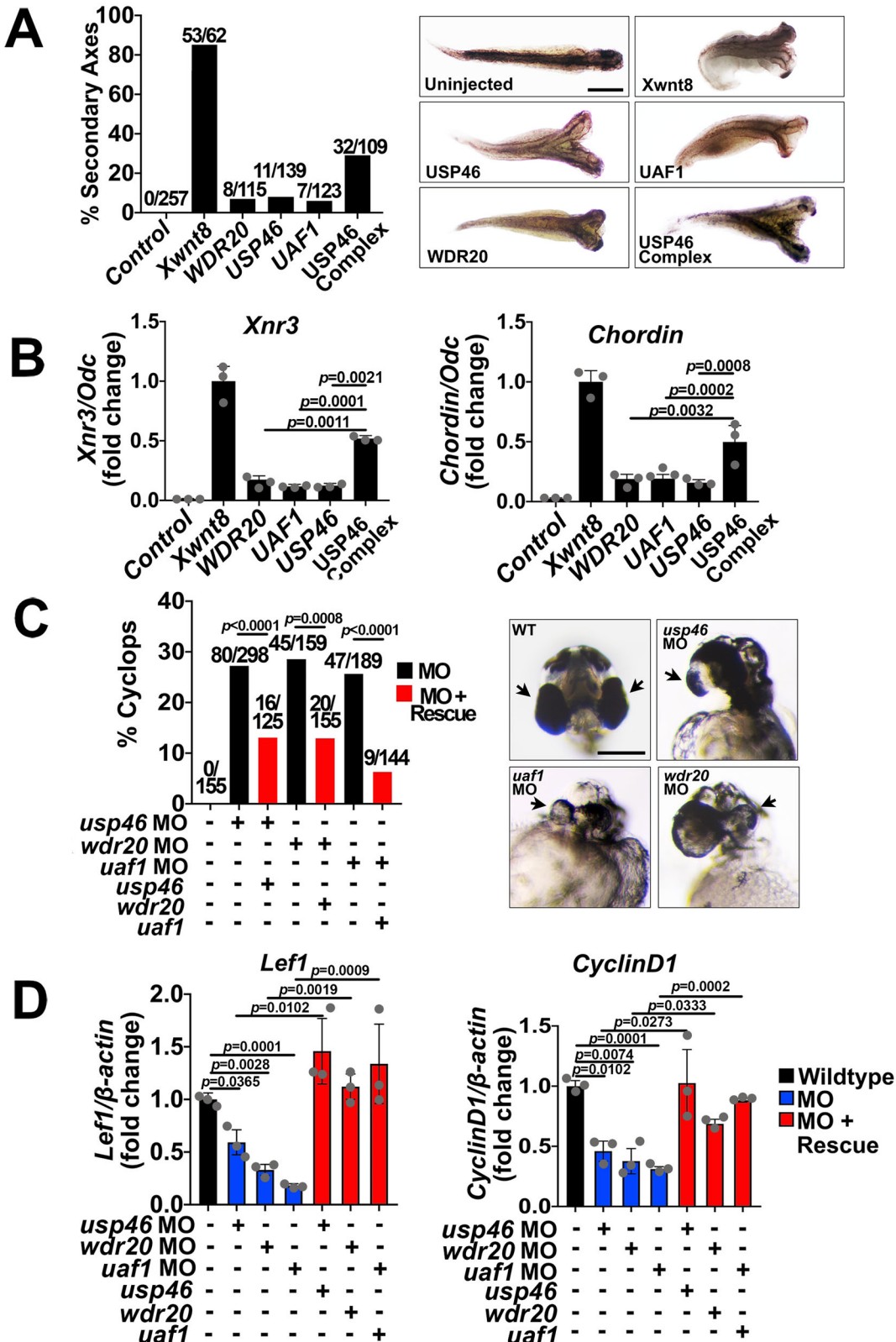

a decrease in LRP6 levels upon knockdown of USP46 in the colorectal cancer cell line, DLD1 (Supplementary Fig. 4C). The decrease in LRP6 levels with USP46 knockdown was not due to reduced *LRP6* mRNA expression (Supplementary Fig. 1A). Similar to USP46 siRNA knockdown, CRISPR-cas9 mediated knockout of USP46 also resulted in decreased LRP6 levels by day six post-treatment (Supplementary Fig. 1C). Thus, the USP46 complex regulates steady-state levels of LRP6 protein.

## The binding of the USP46 complex to LRP6 is enhanced by Wnt stimulation

Given its effect on LRP6 levels, we explored whether the USP46 complex interacted with LRP6. We found that LRP6 co-immunoprecipitated with the USP46 complex using either tagged UAF1 and/or WDR20 (Fig. 3E and Supplementary 5A) and that co-immunoprecipitation of LRP6 was significantly enhanced in the

**Fig. 2 | The USP46 complex regulates Wnt signaling in *Xenopus* and zebrafish.**
**A**, **B** The USP46 complex induces secondary axis formation and Wnt target gene expression in *Xenopus* embryos. **A** *Xenopus* embryos were injected ventrally at the 4-cell stage with *Xwnt8* mRNA (10 pg), individual *USP46* members (1 ng), or a 1:1:1 mixture (0.33 ng each). The percentage of axis duplication is shown with absolute numbers on the top of each bar and representative images. p-values for injection of individual components versus the complex and for injected embryos compared to wildtype is <0.0001. Scale bar = 200 mm. **B** RT-PCR show induction of *Xenopus* Wnt target genes, *Xnr3* and *Chordin*. Expression is shown as a ratio of *Ornithine decarboxylase (Odc)* normalized to *Xwnt8* injected animal caps. Control is un-injected. p-values compare injections of individual components versus the USP46 complex. **C**, **D** Knockdown of the USP46 complex in zebrafish induces a cyclopic phenotype and reduction of Wnt target gene transcripts, which are rescued by co-injecting

homologous human mRNAs. **C** Embryos were injected at the single-cell stage with Morpholino oligonucleotides (MO, 3 ng) and rescued with mRNAs (0.8 ng). The percentage of cyclopic embryos is shown with absolute numbers on the top of each bar. Arrows show developing eyes. p-values compare MO-injected versus rescue. p-values for all injected embryos compared to wildtype is <0.0001. Scale bar = 200 mm. **D** mRNAs were isolated, and *Lef1* and *CyclinD1* levels were quantified by RT-PCR. Gene expression is graphed as a ratio to *β-actin* control and normalized to un-injected embryos. p-values shown compare MO-injected versus un-injected embryos and morpholino-injected versus corresponding rescued embryos. **B**, **D** Graphs show mean ± SEM, n = 3 independent pools of embryos. Significance was analyzed by two-tailed Student's *t* test. Source data are provided as a Source data file.

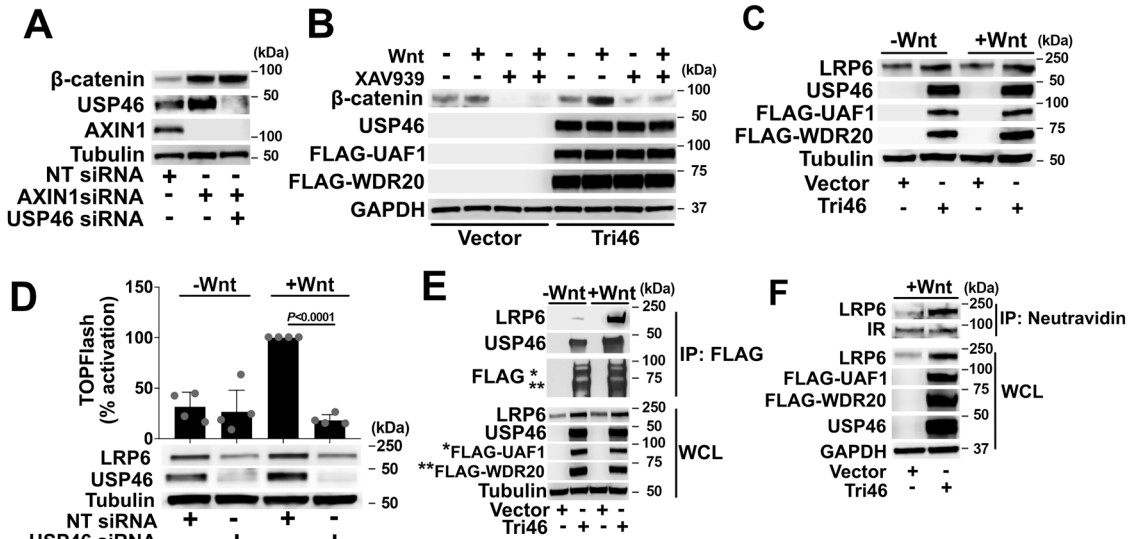

**Fig. 3 | The USP46 complex acts at the level of the Wnt receptor, LRP6.**
**A**, **B** USP46 is upstream of the β-catenin destruction complex. **A** USP46 depletion does not block the stabilization of β-catenin upon *AXIN1* siRNA knockdown. HEK293 cells were transfected with *USP46* and *AXIN1* siRNAs as indicated and immunoblotted for β-catenin. Immunoblotting for AXIN1 and USP46 confirmed their knockdown. **B** The tankyrase inhibitor, XAV939, which stabilizes AXIN to promote β-catenin degradation, inhibits β-catenin stabilization mediated by the USP46 complex. Cells were transfected and treated with Wnt3a in the absence or presence of XAV939 (1 μM) as indicated and immunoblotted for β-catenin, USP46, FLAG-UAF1, and FLAG-WDR20. **C**, **D** Overexpression and knockdown of the USP46 complex increases and decreases steady-state levels of LRP6, respectively. Cells were transfected as indicated, incubated in the absence or presence of Wnt3a, and immunoblotted for LRP6. **C** HEK293 cells were transfected with the USP46 complex (Tri46) in the presence of Wnt3a. **D** HEK293 cells were transfected with non-targeting (NT) or *USP46* siRNAs. Graph shows mean ± SD of TOPFlash normalized to

non-targeting control in the presence of Wnt. p-values compare cells incubated with NT versus *USP46* siRNA in the presence of Wnt3a. Significance was analyzed by two-tailed Student's *t*-test. The graph shows a representative of n = 3 biologically independent experiments. **E** Wnt signaling promotes the association between the USP46 complex and endogenous LRP6. HEK293 cells were transfected with USP46 complex components and treated with Wnt3a as indicated, FLAG-UAF1 (*) and FLAG-WDR20 (**) immunoprecipitated (IP) with anti-FLAG conjugated beads, and co-immunoprecipitated LRP6 detected by immunoblotting. WCL whole cell lysates. **F** The USP46 complex increases cell-surface levels of LRP6. HEK293 cells were transfected with USP46 complex components and treated with Wnt3a, as indicated. Cells were then surface biotinylated, lysates subjected to neutravidin-pulldown, and immunoblotted for endogenous LRP6 and insulin (IR, control) receptors. WCL whole cell lysates. Tubulin and GAPDH are loading controls. All immunoblots are representative of at least three independent experiments. Source data are provided as a Source data file.

presence of Wnt3a. We were unable to perform USP46 co-immunoprecipitation due to the limitations of our anti-USP46 antibody and the fact that tagging USP46 at its amino or carboxyl termini inactivated the protein in our Wnt assays.

To determine whether the USP46 complex acts on the pool of LRP6 at the plasma membrane, we performed a cell-surface biotinylation assay. In this assay, cell surface proteins were labeled with biotin and affinity purified with Neutravidin or Streptavidin bound to beads. We observed that expression of the USP46 complex increased the cell-surface levels of LRP6 but not the insulin receptor control (Fig. 3F). To facilitate detection of endogenous LRP6, we generated a HEK293 cell line, LF203, in which a FLAG tag was knocked into the C-terminal end of LRP6 by CRISPR-Cas9-mediated editing (Supplementary Fig. 5B–D). Consistent with a model in which the USP46 complex acts on the plasma membrane pool of LRP6, we found that USP46, UAF1, and

WDR20 co-immunoprecipitated with cell-surface proteins in the biotinylation assay in both LF203 cells (Supplementary Fig. 5E) and parental HEK293 cells (Supplementary Fig. 5F). These results show that Wnt signaling facilitates the recruitment of the USP46 complex to the pool of LRP6 at the cell surface.

## Wnt signaling induces the assembly of USP46, UAF1, and WDR20 into large cytoplasmic complexes but not into LRP6 signalosomes

We performed gel filtration studies on the cytosolic fraction from HEK293 cells to assess USP46 complex formation in the absence and presence of Wnt signaling. In the absence of Wnt stimulation, USP46, UAF1, and WDR20 migrated in a peak corresponding to the predicted size (~150 kDa) of a globular trimeric complex (Fig. 4A). However, in the presence of Wnt stimulation, the peak was broadened in a manner

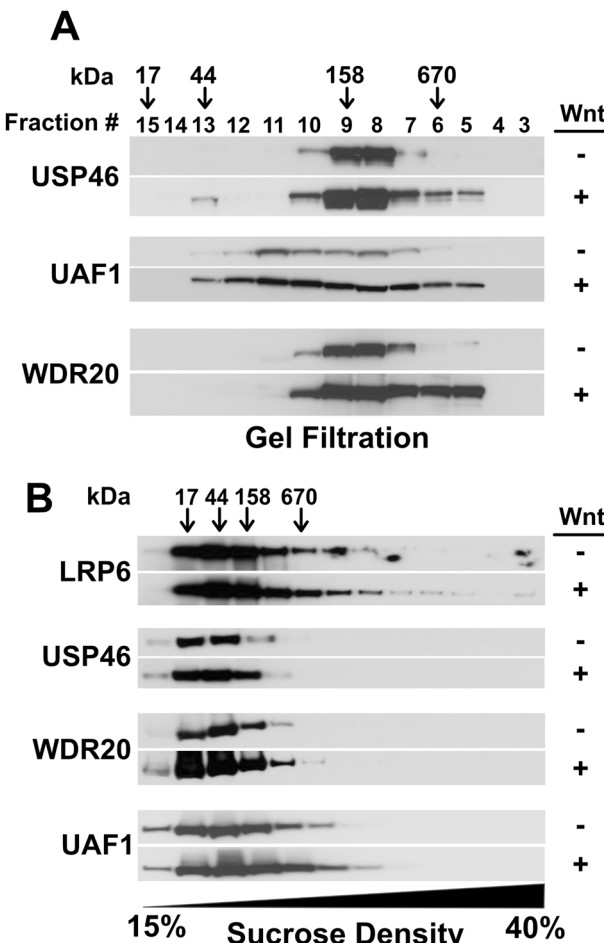

**Fig. 4 | The USP46 complex associates with large complexes independent of LRP6 signalosomes in the presence of Wnt. A** Wnt signaling induces the formation of high molecular weight USP46 complexes as assessed by size exclusion chromatography. HEK293 cells were treated in the absence or presence of Wnt3a, lysates prepared, and high-speed (100,000 × *g*) supernatants passed over a Superdex 200 FPLC column. Fractions obtained were subjected to TCA precipitation followed by immunoblotting. **B** The USP46 complex does not co-fractionate with high molecular weight LRP6 aggregates (signalosomes) on sucrose gradient centrifugation. HEK293 cells were incubated in the absence or presence of Wnt3a, Triton X-100 lysates were collected, and sucrose density gradient (15–40%) sedimentation was performed. Fractions were precipitated with chloroform-methanol extraction and analyzed by immunoblotting. The predicted kDa is based on the elution profile of a set of protein standards. Immunoblots show a representative of three independent experiments. Source data are provided as a Source data file.

consistent with the formation of larger molecular weight complex(es) or a significant change in shape. Thus, Wnt signaling induced a physical alteration of the USP46 complex in the cytoplasm.

Because the association of the USP46 complex with LRP6 was enhanced in the presence of Wnt stimulation (Fig. 3E, F), we tested whether the USP46 complex interacted with LRP6 receptor aggregates in signalosomes that could be readily detected in sucrose density gradients[4]. Signalosomes contain activated Wnt receptors and are critical for the stabilization of β-catenin[4]. HEK293 cells treated with Wnt3a showed a shift in the migration of LRP6 into the denser sucrose density fractions of lysates compared to untreated cells (Fig. 4B). The migration of USP46 did not change, however; it remained in the lighter fractions, even in the presence of Wnt3a. Based on these findings, we conclude that the USP46 complex associates with activated LRP6 receptors but not LRP6 signalosome aggregates.

## USP46 inhibits the ubiquitylation of LRP6

The enzymatic activity of USP46 is required for Wnt activity as a catalytically inactive version of USP46, USP46[C44S], did not increase LRP6 levels (Fig. 5A) or potentiate Wnt signaling in HEK293 STF cells (Supplementary Fig. 6A). The decreases in LRP6 levels and Wnt reporter activity suggest that USP46 [C44S] may be acting in a dominant-negative fashion. To ensure that the differences in the Wnt activity of wildtype USP46 and USP46[C44S] were not due to differences in expression levels, we performed a titration study. We found that, in contrast to wildtype USP46, USP46[C44S] was inactive at all of the concentrations tested (Supplementary Fig. 6B). We similarly found that a mutant of UAF1, UAF1[S170Y], which disrupts its interaction with USP46[17], also failed to increase LRP6 levels (Fig. 5B) and exhibited reduced Wnt activity (Supplementary Fig. 6C). Thus, the enzymatic activity of the USP46 complex is required for its regulation of LRP6 levels.

We next examined if the USP46 complex increases LRP6 levels by promoting its deubiquitylation and opposing the activity of RNF43, an E3 ligase for Wnt receptors. We found that expression of RNF43 in HEK293 and LF203 cells increased ubiquitylation of LRP6 and LRP6-FLAG, respectively, an effect that was blocked by the USP46 complex (Fig. 5C, D). Expression of the RNF43 homolog, ZNRF3, similarly increased ubiquitylation of LRP6-FLAG in LF203 cells and was also blocked by expression of the USP46 complex (Supplementary Fig. 6D). Conversely, knockdown of USP46 in LF203 cells led to increased Ni-NTA pulldown of ubiquitylated LRP6-FLAG (Fig. 5E and Supplementary 6E). Wnt receptors are degraded via the lysosomal pathway upon their polyubiquitylation[11,13], and the lysosome inhibitor, bafilomycin A, was shown to block ubiquitin-mediated degradation of LRP6[33]. Similarly, we found that treatment with Bafilomycin A blocked the decrease in LRP6 levels following knockdown of USP46 in LF203 cells (Supplementary Fig. 6F).

These results indicate that the USP46 complex promotes the deubiquitylation of LRP6 at the plasma membrane, thereby increasing its stabilization and opposing the action of the Wnt receptor ubiquitin ligases, RNF43 and ZNRF.

## USP46 and UAF1 promote the growth of intestinal organoids

Intestinal organoids rely on Wnt signaling for their growth and represent a powerful ex vivo model to study the modulation of the Wnt pathway in a physiologically relevant context[35]. RSPOs, which increase levels of Frizzled and LRP6 by inhibiting RNF43 and ZNRF3 activities, are critical factors required for the culture of intestinal organoids[6,7,36,37]. Because the USP46 complex opposes the action of RNF43 and ZNRF3, loss of USP46 would be expected to potentiate the activities of RNF43 and ZNRF3, decreasing the effectiveness of RSPO in maintaining intestinal organoid growth. We found that knockdown of USP46 decreased the viability of intestinal organoids and LRP6 and β-catenin levels (Fig. 5F and Supplementary Fig. 7A, B) and that this effect was more evident at lower concentrations of RSPO (corresponds to higher RNF43/ZNRF3 activities). Knocking down of UAF1 also decreased the viability of intestinal organoids, and this effect was also greater at lower concentrations of RSPO (Supplementary Fig. 7C–F). Finally, we show that knockdown of both USP46 and UAF1 had a greater effect on the viability of intestinal organoids compared to knocking down USP46 or UAF1 alone (Supplementary Fig. 7D, E).

## The USP46 complex is expressed in the human intestine and is altered in human cancers

Wnt signaling is required for maintaining intestinal homeostasis in adult animals[38]. In order to determine whether the USP46 complex plays a role in Wnt signaling in the intestine, we performed multiplex immunofluorescence on formalin-fixed, paraffin-embedded (FFPE) human intestinal tissue sections (Supplementary Fig. 8). Consistent with a role for the USP46 complex in Wnt signaling in the intestine, we found that all three components are expressed in the small intestine

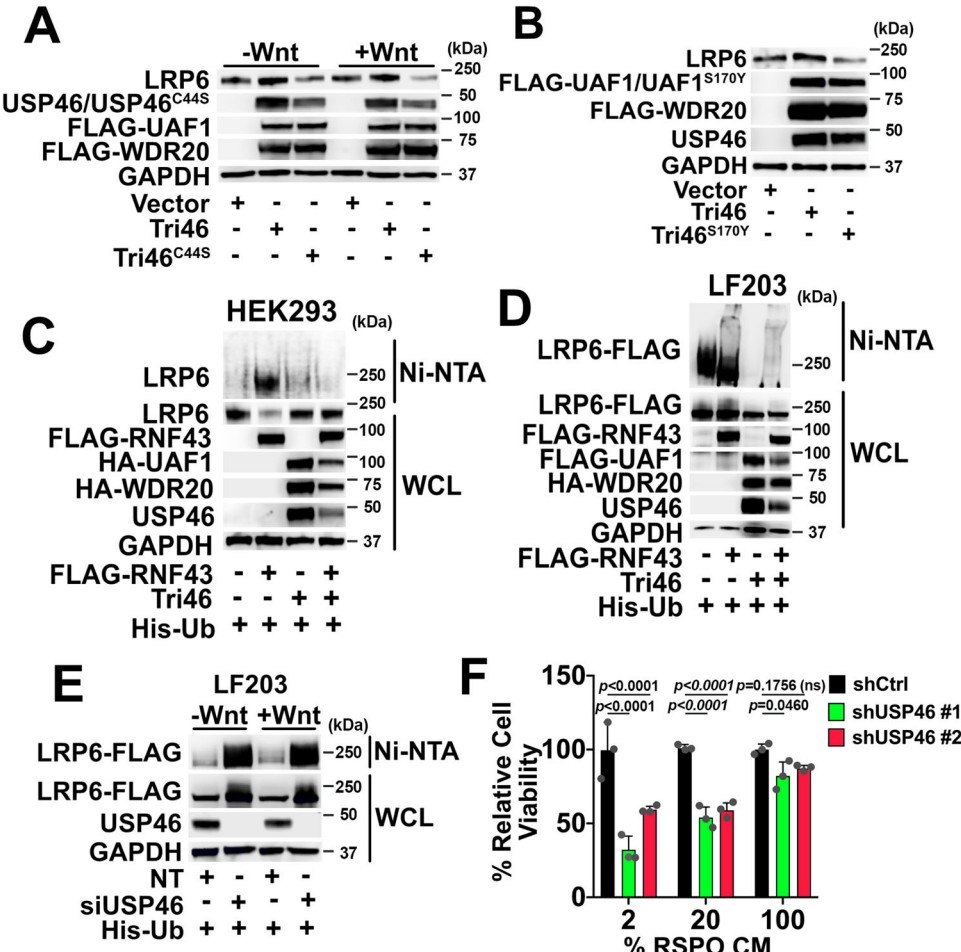

**Fig. 5 | The USP46 complex deubiquitinates LRP6 and opposes the action of the Wnt receptor E3 ligase, RNF43. A, B** The catalytic activity of USP46 is required for its regulation of LRP6 levels. Cells expressing the wild-type or mutant USP46 complexes were incubated in the absence or presence of Wnt3a, and immunoblotting was performed. **A** Overexpression of the USP46 complex containing the catalytically dead USP46[C44S] (Tri46[C44S]) does not increase LRP6 levels. **B** Overexpression of the USP46 complex containing the USP46 binding mutant, FLAG-UAF1[SI70Y], does not increase LRP6 levels. GAPDH is loading control. **C–E** His-ubiquitylation assays. HEK293 cells were transfected as indicated, lysed under denaturing conditions, and His-Ub (hexahistidine-tagged ubiquitin) modified proteins isolated by nickel affinity purification. LRP6 and FLAG-tagged proteins were detected by immunoblotting with anti-LRP6 and anti-Flag antibodies, respectively. **C** Overexpression of FLAG-RNF43 promotes endogenous ubiquitylation of LRP6, which is opposed by overexpression of the USP46 complex. **D** Overexpression of FLAG-RNF43 in LF203 cells similarly promotes LRP6-FLAG ubiquitylation, which is opposed by overexpression of the USP46 complex. **E** Knockdown of USP46 enhances LRP6-FLAG ubiquitination. WCL = whole cell lysates. **F** The sensitivity of intestinal organoids to USP46 depletion is RSPO-dependent. Exogenous RSPO is essential for culturing intestinal organoids in vitro. Intestinal organoids infected with a control lentivirus or two independent lentiviruses expressing USP46 shRNAs were grown with decreasing RSPO conditioned media (CM). Viability was assessed by Cell-Titer Glo. $p \geq 0.05$ is not significant (ns). Significance was analyzed by one-way ANOVA followed by Tukey's multiple comparisons test. Graphs show mean ± SD ($n = 3$ independent experiments). GAPDH is loading control. All immunoblots are representative of at least three independent experiments. Source data are provided as a Source data file.

and colon. The staining of USP46 and UAF1 appear to be more concentrated in the crypts, whereas WDR20 exhibits a more uniform distribution.

Given the role of Wnt signaling as a major driver of tissue growth, we asked whether USP46 is altered in human cancers. Using the cBioPortal to assess 9,125 tumor samples across 33 cancer types in The Cancer Genome Atlas[39,40], we found a significant correlation between USP46 gene alterations and decreased overall cancer patient survival (Supplementary Fig. 9A). High levels of USP46 correlated with reduced survival of patients with cancers such as invasive breast cancer (Supplementary Fig. 9B). Alterations in USP46 mostly consisted of amplification and were commonly observed in glioblastoma multiforme and lung squamous cell carcinomas (Supplementary Fig. 9C).

USP46 is highly expressed in the brain[41], and an analysis of the Cancer Genome Atlas (TCGA)[40] showed that the expression of USP46 correlated (albeit moderately) with both WDR20 and UAF1 in glioblastoma (Supplementary Fig. 10A). Misregulation of the Wnt pathway

has been proposed to contribute to glioblastoma development[42]. We asked if there is a difference in Wnt target gene expression in glioblastoma tumors stratified by expression of the components of the USP46 complex. When comparing triple-high and triple-low USP46/WDR48/WDR20 groups, we observed significantly higher expression of the Wnt target genes, *Nkd1* and *Axin2*, in the triple-high group (Supplementary Fig. 10B). Given these results from analyses of TCGA data, we performed USP46 knockdowns in the glioblastoma cell lines, A172 and U87, to determine whether the complex is active in the glioblastoma cell lines and found significant decreases in LRP6 levels in both lines (Supplementary Fig. 10C).

## Discussion

In this study, we identify the USP46 deubiquitylase complex as an evolutionarily conserved positive regulator of Wnt signaling in vertebrates. Full activity of the USP46 complex in Wnt signaling requires all three components: USP46, UAF1, and WDR20. We show that this

trimeric complex maintains the steady-state level of LRP6 by inhibiting its turnover (opposes the action of the ubiquitin ligases, RNF43 and ZNRF3), thereby increasing cell surface levels of LRP6 (Fig. 6A). Consistent with our data demonstrating that USP46 regulated LRP6 levels, USP46 and UAF1 were identified in a genome-wide screen in a group of novel genes involved in network rewiring at the level of (or above) the Wnt receptors[43].

We found that the USP46 complex is recruited to LRP6 and that knockdown of USP46 does not result in further decreases in LRP6 levels upon Wnt stimulation. Ubiquitylation of Disheveled (Dvl), a positive regulator of the Wnt pathway required for signalosome formation, was found to block its oligomerization in vitro, and it was proposed that deubiquitylation of Dvl is required to promote its polymerization and its subsequent role in signalosome formation[44]. We speculate that similarly, activated, ubiquitylated LRP6 cannot be assembled into signalosomes (possibly due to steric hindrance). Based on our sucrose density gradient experiments, the USP46 complex does not associate with signalosomes, although we cannot rule out low affinity, transient interactions. Thus, Wnt ligand-stimulated recruitment of the USP46 complex may represent a pruning step to maximally increase the availability of active LRP6 receptors for signalosome assembly (Fig. 6B). In an accompanying paper[45], we describe the characterization of the USP46 complex in Drosophila, which provides in vivo evidence for an evolutionarily conserved role of the USP46 complex in regulating Arrow/LRP6 levels during Wingless/Wnt signaling.

How the recruitment of the USP46 complex to LRP6 occurs in a Wnt-dependent manner is not clear. Dvl recruits RNF43 and ZNR3 to Wnt receptors, promoting their degradation[46]. We have been unsuccessful in demonstrating that Dvl also plays a role in the recruitment of USP46 to LRP6. It remains to be determined whether the USP46 complex binds LRP6 as a pre-formed trimer or whether it proceeds through stepwise recruitment of individual subunits. In addition, based on our gel filtration experiments, Wnt-dependent recruitment may involve the formation of a larger molecular weight cytoplasmic complex (Fig. 6B).

Currently, no FDA-approved drugs target the Wnt pathway. To our knowledge, there are no reported small molecules that inhibit the USP46 complex. We anticipate that inhibitors that block USP46 complex activity could either block its catalytic site or promote complex dissociation and, subsequently, protein turnover. Wnt-driven cancers

that could be targeted by USP46 inhibitors include invasive breast cancer, glioblastomas, and colorectal cancer (CRC). In the latter case, susceptible CRCs may include those primarily driven by Wnt receptor activity (e.g., mutations in RNF43, ZNF3, and RSPO). A recent study suggests that CRCs with APC mutations (~80 of all nonsporadic cases) may also be driven, in part, by Wnt receptor activation[47]. Thus, inhibitors that target USP46 may have broad applicability for treating Wnt-driven human cancers.

## Methods

### Cell lines
HEK293 (CRL-1573), HEK293STF (CRL-3249), DLD-1 (CCL-221) were purchased from the American Type Culture Collection (ATCC), except A172 and U87, which were from the William Weiss Lab (UCSF). Cells were grown in DMEM supplemented with 8% FBS except for DLD1 cells, which were grown in RPMI supplemented with 8% FBS. All cell lines were tested negative for Mycoplasma contamination.

### Generation of the LF203 cell line
The gRNA sequence used for targeting the C-terminus of LRP6 was identified using the CHOPCHOP web tool. Briefly, the LRP6 gRNA was cloned into the pCas-Guide-EF1a-GFP vector (Origene) following the manufacturer's protocol. HEK293 cells were transfected with the LRP6 gRNA plasmid and a gene block (IDT) encoding homology arms flanking a FLAG tag using Lipofectamine 3000. After 48 h, cells were single-cell sorted (BD FACSMelody) into 96-well plates and grown to confluency. Positives were identified by immunoblotting for the FLAG epitope. Genomic DNA was collected from the parental line and the LF203 cell line using the Wizard Genomic DNA Purification Kit (Promega). Genomic DNA was submitted to the Vanderbilt VANTAGE Core for whole genome sequencing to verify the integration and positioning of the FLAG tag on the C-terminus of LRP6.

### Antibodies for immunoblotting
The following antibodies were used for immunoblotting: Rat anti-HA (Roche, ROAHAHA), Rabbit anti-FLAG (Proteintech, 20543-1-AP), Mouse anti-GAPDH (Developmental Studies Hybridoma Bank, hGAPDH-2G7), Rabbit anti-USP46 (Proteintech, 13502-1-AP), Mouse anti-Tubulin (Developmental Studies Hybridoma Bank, E7), Mouse anti-β-catenin (BD Transduction Laboratory, 610154), Rabbit anti-Axin1 (Cell Signaling Technology, 2087), Rabbit anti-Insulin Receptor β (Cell

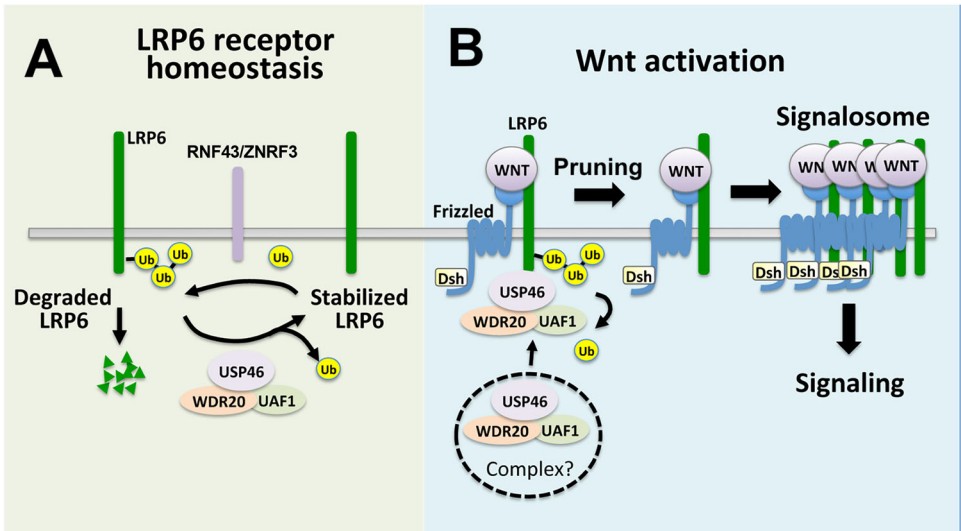

**Fig. 6 | Model of USP46 complex regulation of LRP6 in the Wnt pathway. A** The USP46 complex catalyzes the deubiquitylation of LPR6, opposing the activities of RNF34 and ZNFR3 (and potentially other E3 ligases) to maintain low homeostatic levels of LRP6. **B** Upon Wnt ligand binding to Frizzled and LPR6, the USP46 complex is recruited to and deubiquitylates LPR6, allowing LRP6 to be assembled into the Wnt signalosome.

Signaling Technology, 3025), Rabbit anti-LRP6 (Cell Signaling Technology, 2560), Rabbit anti-WDR20 (Bethyl Laboratories, A301-657A), Rabbit anti-WDR48 (UAF1) (Proteintech, 16503-1-AP), Goat anti-rat IgG H + L-HRP (Thermo, 31470), Goat anti-mouse IgG H + L-HRP (Promega, W4021), Goat anti-rabbit IgG H + L-HRP (Promega, W4011). All primary antibodies were used at 1:1000 dilution except anti-FLAG (1:2000), anti-GAPDH (1:500), and anti-WDR20 (1:2500). All secondary HRP antibodies were used at 1:5000 dilution.

## Generation of CRISPR knockouts

To generate USP46 KO, UAF1 KO, or WDR20 KO cell lines, HEK293 Cas9 cells were transfected with 50 nM of two independent crRNAs targeting *USP46*, *UAF1*, or *WDR20* and 50 nM tracrRNA (#U-0020005-20, Horizon Discovery). Transfections were performed using Dharmafect1 (Horizon Discovery). The crRNA and tracrRNA were purchased from Horizon Discovery, and target sequences are as follows:

*USP46* (CM-006092-01-0002): TAATAAACCAGAACTCACCT
*USP46* (CM-006092-03-0002): GTGATATAATGCCCACGATT
*UAF1* (CM-016462-01-0002): ATCACCGGCAGAACACAGCA
*UAF1* (CM-016462-03-0002): TATGGAACACCATACTGATT
*WDR20* (CM-021318-02-0002): CTGGTCGTTGAGGTTTACGA
*WDR20* (CM-021318-04-0002): GCTTTCCAAGATGGCGACGG

## Transfections and RNAi-mediated knockdowns

Plasmid transfections were performed using the calcium phosphate method unless otherwise noted. siRNA transfections were performed using Dharmafect1 (Horizon Discovery) following the manufacturer's protocol. Knockdowns with single siRNAs or pooled siRNAs were performed at a final concentration of 50 nM. Sequences for siRNAs (5'->3') used are as follows:

*USP46*:

siRNA #1 F: UCCGAAACAUCGCCUCCAUUU, R: AUGGAGGCGAUG UUUCGGAUU

siRNA #2 F: GCAGGAUGCUCAUGAAUUUUU, R: AAAUUCAUGAGC AUCCUGCUU

*UAF1*:

siRNA #1 F: CAGCAGAGAUGUAUAGCAAUU, R: UUGCUAUACAUC UCUGCUGUU

siRNA #2 F: CCUAAGAAACCCUGACAUUUU, R: AAUGUCAGGGUU UCUUAGGUU

*Axin1*:

siRNA #1 F: GCGUGGAGCCUCAGAAGUUUU, R: AACUUCUGAGGC UCCACGCUU

siRNA #2 F: CCGAGGAGAAGCUGGAGGAUU, R: UCCUCCAGCUUC UCCUCGGUU

## RNA isolation and qRT-PCR

HEK293 STF cells were treated with siRNA for 24 h followed by incubation with Wnt3a (10 ng/ml, Time Biosciences) for 8 h. Total RNA was isolated using RNAeasy RNA extraction kit (Qiagen) and reversed transcribed to cDNA using the High-Capacity cDNA Reverse Transcription Kit (Applied Biosystems). qPCR assays were performed in quadruplicate using TaqMan FAST Advanced Real-time PCR Master Mix (Applied Biosystems) and analyzed on a CFX96 Real-Time PCR system (Bio-Rad). Predesigned and revalidated TaqMan probes (Life Technologies) for the genes and Assay ID used were as follows: DKK (Hs00183740_m1), *LRP6* (Hs00233945_m1), *GUSB* (Hs00939627_m1), *LEF1* (Hs01547250_m1), and SP5 (Hs01370227_mH).

For *Xenopus* studies, RNAs were prepared from animal cap explants of stage 10.5 embryos (11 h post fertilization at 23 °C). For zebrafish, RNAs were prepared 24 h post fertilization (24 hpf at 28.5 °C). Samples were homogenized in 1 ml RNA Stat-60 (Amsbio) with a disposable pestle and extracted with chloroform. All qPCR amplifications were performed in biological and technical triplicate. PCR primers (5'->3') used are as follows:

*Odc*: F: GTCAATGATGGAGTGTATGGATC, R: TCCATTCCGCTCT CCTGAGCAC
*Chordin*: F: AACTGCCAGGACTGGATGGT, R: GGCAGGATTTAGA GTTGCTTC
*Xnr3*: F: CTTCTGCACTAGATTCTG, R: CAGCTTCTGGCCAAGACT
*β-actin:* F: CGAGCTGTCTTCCCATCCA, R: CACCAACGTAGCTG TCTTTCTG
*CyclinD1:* F: GCCAAACTGCCTATACATCAG, R: TGTCGGTGCTTTT CAGGTAC
*Lef1*: F: GAGGGAAAAGATCCAGGAAC, R: AGGTTGAGAAGTCTAG CAGG

## Reporter assays

For cell-based luciferase TOPFlash assays with HEK293 STF cells, cells were incubated with 10 ng/ml purified Wnt3a (Time Bioscience) for 24 h prior to lysis with 1× Passive Lysis buffer (Biotium) on a shaker for 15 min. From the lysates, samples were taken to measure luciferase activity using the Steady-Glo Luciferase Assay (Promega) and cell viability using the Cell-Titer Glo Assay (Promega). Luciferase signals were normalized to cell viability Cell-Titer Glo signals. For Dual-Glo (Promega) assays, lysates were prepared as above. Firefly and Renilla luciferase signal measurements were performed according to the manufacturer's instructions. Firefly luciferase signals were normalized to co-transfected Renilla luciferase signals. Luminescence was detected with a FLUOstar Luminometer (Optima). Assays were performed in either triplicates or quadruplicates and repeated at least 3 times.

## Preparations of samples for immunoblotting

For whole-cell lysates, cells were obtained using non-denaturing lysis buffer (NDLB) (50 mM Tris-HCl pH 7.4, 300 mM NaCl, 5 mM EDTA, and 1% Triton X-100 (w/v) supplemented with 1 mM PMSF, 1× PhosSTOP inhibitor (Roche), and 10 mM NaF). Samples were gently agitated at 4 °C for 30 min, followed by clarification by spinning in a microfuge at 13,000 RPM for 10 min at 4 °C. For cytoplasmic fractions, cells were lysed using 10 mM HEPES pH 7.8, 10 mM KCl, 2 mM MgCl$_2$, 0.1 mM EDTA, 1 mM PMSF, 1× PhosSTOP inhibitor (Roche), and 10 mM NaF, and incubated on ice for 30 min. Lysates were vortexed, sheared 8 times with a 25 G needle, and clarified by spinning in a microfuge at 13,000 RPM for 5 min at 4 °C. Proteins were analyzed by SDS-PAGE and immunoblotting. Chemiluminescence signal was detected using a C-DiGit blot scanner (LI-COR). Obtained images and band intensity were analyzed using Image Studio (LI-COR).

## Co-immunoprecipitation studies

Cells were lysed in NDLB, and lysates were diluted to 1 mg/ml with NDLB and incubated with antibodies overnight at 4 °C. Samples were then incubated with Protein A/G magnetic beads (Millipore) or anti-FLAG agarose beads for 2 h (Sigma). Beads were washed 3× with NDLB, and sample buffer was added to elute the bound protein (37 °C for 1 h).

## Cell-surface biotinylation assays

HEK293 cells were prepared by washing 3× with pre-chilled modified PBS (1X PBS supplemented with 0.9 mM CaCl$_2$ and 0.5 mM MgCl$_2$) on ice. Biotinylation was initiated by incubating the cells with 0.5 mg/ml EZ-Link Sulfo-NHS-SS-Biotin (Thermo Fisher Scientific) in modified PBS with gentle rocking for 30 min at 4 °C. The reaction was then quenched by washing the cells 2× with 50 mM Tris-HCl (pH 7.4) for 10 min. Whole-cell lysates were then collected with RIPA buffer (50 mM Tris-HCl pH 7.4, 150 mM NaCl, 1% Triton X-100, 0.5% sodium deoxycholate, 0.1% SDS, 1 mM EDTA) supplemented with 1 mM PMSF, 10 mM NaF, and 1× PhosSTOP. Lysates were gently agitated for 30 min at 4 °C and clarified by spinning at 16,000 × g in a microcentrifuge for 10 min at 4 °C. Biotinylated proteins were purified with NeutrAvidin agarose beads (Pierce). Samples were analyzed by SDS-PAGE, followed by immunoblotting.

## Ubiquitylation assays

Cells were collected for ubiquitylation assays as described previously[23]. Briefly, cells were spun down for 30 s at $10,000 \times g$ in a microcentrifuge at 4 °C, washed in ice-cold 1× PBS, and lysed in Buffer A (6 M guanidine-HCl, 0.1 M $Na_2HPO_4/NaH_2PO_4$ (pH 8.0), and 10 mM imidazole). Lysates were sonicated at max power (8× at 1 s intervals), added to Ni-NTA agarose beads (Qiagen), and incubated for 3 h at RT with rotation. Beads were washed with 2× Buffer A, 2× Buffer A/TI (25 mM Tris-HCl (pH 6.8), and 20 mM imidazole, 1:3 v/v), and 1× Buffer TI. Bound samples were eluted with sample buffer supplemented with 0.2 M imidazole, followed by SDS-PAGE and immunoblotting.

## Organoid cultures

Wild-type mouse intestinal organoids were isolated and cultured as previously described[48]. Prior to infection, organoids were collected and dissociated by incubating in Gentle Cell Dissociation Reagent (StemCell Technologies) for 10 min at RT. For infection, 7500 dissociated organoids were resuspended in a mixture containing lentivirus (MOI = 10), 8 µg/ml polybrene, and 10 µM Y27632 (StemCell Technologies) in 25% L-WRN conditioned media. The organoid-lentivirus mixtures were then spinoculated at $600 \times g$ for 2 h at RT, followed by incubation at 37 °C for 4 h. After incubation, organoids were pelleted, washed with Intesticult growth media (StemCell Technologies), resuspended in ice-cold Matrigel (Corning), seeded in a 48-well plate, and overlaid with Intesticult growth media. After 72 h, the percentage of RSPO-conditioned medium in the culture media was adjusted according to experimental design. Images were taken after 9 days using an Olympus IX51 inverted fluorescence microscope and Olympus CellSens software. The viability of organoids was assayed using Cell-Titer Glo (Promega) following the manufacturer's protocol.

$APC^{min}$ organoids were collected and digested with Gentle Cell Dissociation Reagent (Stemcell Technologies) for 10 min at RT, followed by shearing with a 25 G needle once. Dissociated organoids were added to the lentiviral mixture (lentiviral particles (Dharmacon, MOI = 10), 8 µg/ml polybrene, and 10 µM Y27632 (Stemcell Technologies) in 25% L-WRN conditioned media. The organoid-lentivirus mixture was spinoculated at $600 \times g$ for 2 h using a tabletop centrifuge at RT, followed by incubation at 37 °C for 4 h. Infected organoids were resuspended in Matrigel (Stemcell Technologies), plated at 10,000/well in a 48-well plate, and overlaid with culture media. After 9 days, organoids were imaged, collected for RNA extraction, lysed for immunoblotting, and assayed for cell viability. Images were obtained using an Olympus IX51 inverted fluorescence microscope and Olympus CellSens software. The organoid size was measured using the Olympus CellSens software. For cell viability and immunoblotting, organoids were lysed in RIPA buffer (Thermo) supplemented with a Protease/Phosphatase inhibitor cocktail (Thermo) at 4 °C for 10 min. 1/20 of the lysates were used in Cell-Titer Glo assays (Promega) following the product's instructions to measure cell viability. The remaining lysates were boiled in SDS sample buffer for 10 min, followed by SDS-PAGE analysis and immunoblotting.

## Gel filtration

HEK293 cells were lysed in detergent-free NDLB and sheared with a 25 G needle. Lysates were spun at $16,000 \times g$ in a microcentrifuge for 10 min at 4 °C, followed by centrifugation at $100,000 \times g$ for 20 min at 4 °C. Samples (normalized for protein concentration and volume) were loaded onto a Superdex 200 10/300 gel filtration column connected to a BioLogic DuoFlow System (Bio-Rad). Fractions were collected, concentrated by TCA precipitation, and pellets were resuspended in sample buffer for SDS-PAGE and immunoblotting. The following standards were used for calibration: thyroglobulin (670 kDa), γ-globulin (158 kDa), ovalbumin (44 kDa), and myoglobin (17 kDa) (Bio-Rad).

## Sucrose density gradients

Equivalent amounts of whole-cell lysates in 300 µl volumes were gently layered on top of 5 ml 15–40% sucrose gradients (50 mM Tris-HCl pH 7.4, 300 mM NaCl, 0.02% Triton X-100 (w/v), 10 mM NaF, 1× PhosSTOP tablet inhibitor, 1× protease inhibitor cocktail). Gradients were spun at 43,000 RPM for 4 h at 4 °C. Fractions (350 µl) were manually collected from the top of the gradient and concentrated by chloroform-methanol precipitation. Precipitated samples were resuspended in sample buffer and analyzed by SDS-PAGE and immunoblotting.

## Multiplex immunofluorescence of formalin fixed paraffin embedded (FFPE) tissue

De-identified normal human intestinal tissue was obtained through an IRB-approved Exempt Study (IRB #231475). Formalin-fixed paraffin embedded tissue from de-identified subjects was obtained without clinical history or a link back to the original study subject. Unstained slides were collected and used for multiplex immunofluorescence. 5 µm normal colon and small intestine tissue sections were cut from FFPE blocks and stored at −20 °C. Tissue sections were thawed overnight at room temperature and heated for 1 h at 60 °C. Tissue sections were deparaffinized with xylene (2 × 15 min), ethanol (100% 2 × 5 min, 95% 1 × 5 min), and water (5 min) and washed with PBS. Antigen retrieval was performed by heating slides for 45 min in sodium citrate buffer (pH 6.0) in a rice cooker, followed by 30 min at room temperature. Tissues were washed with PBS and blocked for 2 h with 10% goat serum in PBS (blocking buffer). Primary antibodies were diluted in blocking buffer and incubated on tissue sections at 4 °C for 16 h. Tissue sections were washed with 0.05% Tween 20 in PBS. Secondary antibodies were diluted in blocking buffer containing Hoechst 33342 nuclear stain (1:1000) and incubated on tissue sections at 37 °C for 1 h. Images were taken on a Nikon Spinning Disc confocal microscope.

The primary antibodies for immunofluorescence studies of the USP46 complex were the same as those for immunoblotting except they were used at higher concentrations: Rabbit anti-USP46 (1:100), Rabbit anti-WDR20 (1:100), Rabbit anti-WDR48 (UAF1) (1:50), and Mouse anti-β-catenin (1:300). Secondary antibodies used were Goat anti-mouse Cy3 (1:100, Abcam, ab97035) and Goat anti-rabbit Alexa Fluor 647 (1:50, Invitrogen, A-21245).

## Xenopus and zebrafish injections

For injections, capped mRNAs were generated using mMessage mMachine (Ambion) according to the manufacturer's instructions. *Xenopus laevis* embryos were in vitro fertilized, dejellied, cultured, and injected as previously described[49]. Embryos were injected equatorially in the ventral blastomeres at the 4-cell stage and allowed to develop to stage 35 (-2 days post fertilization at 23 °C) before phenotyping. For *Xenopus* embryos, simultaneously injecting mRNAs of all three USP46 components at a 1:1:1 ratio (1 ng total) reliably resulted in duplicated axes. For zebrafish injections, wild-type (NHGRI-1 strain) zygotes (1 cell) were injected (1 nl) and phenotyped at 3 days post fertilization at 28.5 °C. Scoring of the cyclopic phenotype is as described previously[50]. Only phenotypes observed in $n \geq 3$ biological repeats were reported. Embryos with severe and non-specific edema were excluded from the analysis. Embryos from $n \geq 3$ male/female pairs were collected per biological repeat. Morpholinos (MO) with the following sequences (5'->3') were purchased from Gene Tools:

Standard Control: CCTCTTACCTCAGTTACAATTTATA
*WDR20*: CTCCGCCGCCATCTTTGACATTTAC
*UAF1*: GAAGCGTCGCCATCTTGCATGTTG
*USP46*: CGATGTTTCTGACAGTCATTTAGTT

For zebrafish CRISPRi studies, embryos were injected with combinations of 250 pg dCas9 mRNA and 250 pg sgRNA (see Supplementary Table 1). To assess the degree of mRNA downregulation, qPCR of USP46 components was performed on pools of 10 fish collected at 24 hpf using the following primers (5'–3'):

*ups46*

Set #1 F: CCCAAACAGAGGGCATTACA, R: CTATCGCCTGCGCATC TATT

Set #2 F: GCAAACAAGAGGCACAGAAAC, R: CAGTGGAAAGACCA CACGATAA

*wdr48A*

Set #1 F: CAGCGGAAACAAGGACTCTATC, R: TCGTGGATCCCACA CTCTTA

Set #2 F: GAACGCACAAGGACTATGTTAAAG, R: TCACATCCCAGA GGAAGATTTG

*wdr48B*

Set #1 F: CTCTGACCGCCTCCAATAATAC, R: AATGATGACCGT TCCTGTCTG

Set #2 F: CCACACACTCCTGTGATCTTT, R: CCACTGAGGTAC GGTTTCATT

*wdr20A*

Set #1 F: GCTGACCGTGCTCATCTT, R: GACGACAACAAACCC ACTTTC

Set #2 F: GTCAGAGTGTCGTTCGTCAA, R: CTTCCGCACGCCTTTA TAGAT

*wdr20B*

Set #1 F: CGAGAAAGGAGAAGGAGCATAAG, R: GTCTTACACCCGT CCGATTTAC

Set #2 F: CATCTGCCCATCCATCTATTCA, R: ACGGACAGACAG ACAGATAGA

Fish were fixed with 4% PFA at 3 days post fertilization (dpf), evaluated, and imaged. Embryos analyzed were an aggregate from ≥3 clutches of embryos from ≥2 breeding pairs.

## Microscopy of embryos

Bright field images were obtained using a Stemi 2000-CS microscope (Zeiss, Oberkochen, Germany) with an Olympus DP72 camera. Images were analyzed in Fiji or Photoshop.

## Statistics

Data were analyzed with PRISM 9.1 and R Software. Statistical analyses were performed in R v3.1.0. Fisher's exact test and multiple *T* test (two-tailed, equal variance). One-way ANOVA followed by Tukey's multiple comparisons test. Post hoc analysis of Fisher's exact test and multiple *T* test tests were by Bonferroni correction. All experiments were performed at least 3 times and frequently replicated by multiple lab members. For *Xenopus* and zebrafish studies, sample sizes (*n*) are indicated as *n* = number of samples, with the number of the observed phenotype over the total sample number.

## cBioPortal and TCGA data analysis

For glioblastoma, mRNA expression data of USP46, WDR48, WDR20, NKD1, and AXIN2 (obtained using the Agilent-4502A platform) in TCGA-GBM patient tumors was downloaded from Gliovis (http://gliovis.bioinfo.cnio.es/). The high- or low-expression group for a specific gene was divided by median expression level. *NKD1* and *AXIN2* mRNA expression levels were checked in the USP46-high/UAF1-high/WDR20-high group and the USP46-low/UAF1-low/WDR20-low group.

To broadly assess for USP46 gene alterations in human cancers, cBioPortal was used to assess the genetic alterations in the USP46 gene across 9,125 tumor samples across 33 cancer types in The Cancer Genome Atlas. To evaluate USP46 mRNA expression in The Cancer Genome Atlas, we utilized the ULCAN interactive web-portal tool[51].

## Animal care

All animal experiments in this study (*Xenopus* and zebrafish) were performed following the protocols approved by Vanderbilt University's and the University of Maryland School of Medicine's Institutional Animal Care and Use Committees.

## Reporting summary

Further information on research design is available in the Nature Portfolio Reporting Summary linked to this article.

## Data availability

The authors declare that all data supporting the findings of this study are available within the published article and its Supplementary Information files. Raw data and original images are included in the Source data file provided within the manuscript. Further information and requests for resources and reagents should be directed to and will be fulfilled by the lead contact, E.L. (ethan.lee@vanderbilt.edu). Source data are provided with this paper.

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

## Acknowledgements

We thank all members of the Lee and Ahmed laboratories for their insightful advice and discussion regarding this work, Mary Rockouski for technical support, and Anna Schwarzkopf and Carolina Cywiak for their thoughtful comments on the manuscript. This work was supported by the National Institutes of Health (NIH) Grants: T32CA00959228 to V.H.N.; T32HD0075032 and T32AR007592 to L.R.N.; T32GM008554 to S.N.K.; T32CA009582 to C.C.A.; T32GM007347 and F30CA281125 to M.A.L.; R35GM122516 and P50CA236733 to E.L.; R01CA219189 to D.J.R.; R01GM118557 to C.C.H.; R01GM121421, R01GM122222, and R35GM136233 to Y.A.; K08CA240901 and R01CA272875 to V.L.W.; R01GM118082 and R21HD101980 to R.R.; P50CA097257 and P30CA82103 to W.A.W.; and R01CA244188 and R01CA281002 to D.J.R., E.L., and Y.A. Support also came from the American Cancer Society grants 133934-CSDG-19-216-01-TBG and RSG-22-084-01-MM to V.L.W.; the Cancer Research UK Brain Tumour Award (A28592), the Samuel G. Waxman Cancer Research Foundation (A128431) and the Evelyn and Mattie Anderson Chair to W.A.W.; the Intramural Research Program of the National Institutes of Health, National Cancer Institute, and Center for Cancer Research to A.M.L.; the American Heart Association (829471) to J.M.B.; and the Vanderbilt Center for Structural Biology to H.K.K.

## Author contributions

V.H.N. conceptualized the hypothesis, designed experiments, performed experiments, analyzed data, and wrote the manuscript. Z.S. contributed ideas for the hypothesis, performed experiments, and analyzed data. L.R.N., C.W., and C.C.H. designed experiments, performed experiments, and analyzed data for all experiments in *Xenopus* and zebrafish. A.N., C.S., and D.J.R. designed experiments, performed experiments, and analyzed data for all organoid experiments. M.A.L. designed experiments, performed experiments, and imaged sections of FFPE tissue. S.N.K. performed qRT-PCR experiments in cultured cells and analyzed data. H.K.K. helped with the gel filtration experiments. Z.A. and W.A.W. performed the bioinformatic analyses of glioblastoma tumors from TCGA. C.C.A., J.M.B., A.L.L., and L.G. generated and performed CRISPR KO cell experiments. A.G.H. performed initial

experiments that formed the basis for this work. H.B. contributed ideas to support this work. S.D. and J.L. helped with immunoblotting. A.M.L. and R.R. contributed ideas to support this work. V.L.W. performed bioinformatic analyses. Y.A. and E.L. conceptualized the hypothesis, designed experiments, analyzed data, and revised the manuscript.

## Competing interests

E.L., D.J.R., and W.A.W. are co-founders of StemSynergy Therapeutics Inc., a company that seeks to develop inhibitors of major signaling pathways (including the Wnt pathway) for the treatment of cancer. The other authors declare no competing interests.

## Additional information

[1]Department of Cell & Developmental Biology, Vanderbilt University, Nashville, TN 37232, USA. [2]Program in Cancer Biology, Vanderbilt University, Nashville, TN 37232, USA. [3]Department of Molecular and Systems Biology and the Dartmouth Cancer Center, Geisel School of Medicine at Dartmouth College, Hanover, NH 03755, USA. [4]Department of Medicine, University of Maryland School of Medicine, Baltimore, MD 21201, USA. [5]Department of Oncology, Lombardi Comprehensive Cancer Center, Georgetown University, Washington, DC 20057, USA. [6]Department of Pathology, Microbiology, and Immunology, Vanderbilt University Medical Center, Nashville, TN 37232, USA. [7]UCSF Helen Diller Family Comprehensive Cancer Center, San Francisco, CA 94158, USA. [8]Department of Pediatrics, University of California San Francisco, San Francisco, CA 94158, USA. [9]STEMCELL Technologies, 1618 Station Street, Vancouver, BC V6A 1B6, Canada. [10]Laboratory of Cellular and Molecular Biology, Center for Cancer Research, National Cancer Institute, National Institutes of Health, Bethesda, MD 20892, USA. [11]Departments of Biochemistry, Stanford University School of Medicine, Stanford, CA 94305, USA. [12]Vanderbilt Ingram Cancer Center, Vanderbilt University Medical Center, Nashville, TN 37232, USA. [13]These authors contributed equally: Victoria H. Ng, Zachary Spencer.
✉e-mail: yashi.ahmed@dartmouth.edu; ethan.lee@vanderbilt.edu

