## [Peer Review File · Nature Communications]

The USP46 complex deubiquitylates LRP6 to promote Wnt/ β -catenin signalingREVIEWER COMMENTS

Reviewer #1 (Remarks to the Author):

This manuscript brings forth a new and important protein complex that positively enhances WNT signalling at the cell membrane. It is mechanistically compelling with in vivo and in vitro relevance. It lacks analyses at the cellular level as is suggested in our comments below:

Analyzing which WNT-driven human organs express the USP46 complex would help to clarify if this WNT control mechanism is global or specific to individual tissues.

What are the spatial-temporal expression of each components in developing zebrafish embryos since they are shown to be required for A/P axis formation?

Please document their developmental profile .

While we understand that making triple KO in fish (or Xenopus) would be difficult , the use of Morpholinos in these model animals is no longer advised (there are too many off targets effects even when shown to be rescuable). The work by C. Niehrs and colleagues on this important technical shift, also published in Nature Comm earlier (<https://www.nature.com/articles/s41467-020-19373-w>), should encourage the authors to revisit some experiments with the use of F0 crispants in fish.

The study is mostly based on IP and western blot analyses.

Showing the subcellular localization and co-localization of the individual USP46 complex components by IF and other means would strengthen the results considerably.

Likewise making CRISPR/Cas9 ko cells in 293T cells is now very attainable. The paper relies too much on siRNAs.

Figure specific comments:

It seems that the information in figure 1C is entirely duplicated in 1D. In consequence, the authors should remove 1C. In the bar diagram of 1C/D, the value of Wnt3a treated cells in the absence of Tri46 has no error bars. Does this value represent a single read? If so, I would ask to repeat this experiment and include the average of 3 values for the sake of accuracy.

In 1E/F, the authors show the effect of USP46 and UAF1 knockdown in human cells. The data of their attempt to knockdown WDR20 is not shown as the knockdown was not efficient. Does the result at least trend in the right direction? Alternatively, are potentially other USP46 co-factors involved in human cells?

Figure 2a is misleading, there are almost no secondary axes in Wdr20, or Uaf1 or Usp46 (6 out of 101) single injected-embryos compared to control. Why are such low numbers illustrated with duplicated axis images on the right.

Is it statistically significant compared to control (0 out of 159) ? Is the triple o/e assay without effect in a B-cat Morphants? What type of dose curve was tested/established to compare single mRNA vs triple mRNA injections for panels A and B.

Numbers of injected zf embryos in panel C are really too low.

In Figure 5C/D, the authors show that the expression of RNF43 in HEK293 and LF203 cells increased ubiquitylation of LRP6 and that the USP46 complex blocked this effect.

ZNRF3 represents an additional E3 ligase for WNT receptors. Does USP46 equally counteract ZNRF3 induced ubiquitination of LRP6?

Figure 5F represents data analyzing the effect of the USP46 complex in intestinal organoids. In general, we would like to understand better which intestinal cells express all three components of the USP46 complex. Given the WNT gradient along the crypt-villus axis, we would assume the highest USP46 complex levels at the crypt base.

The authors show that knockdown of USP46 decreased the viability of intestinal organoids (Fig. 5F). Does knocking down UAF1 or WRD20 has a similar effect on organoid viability? Given that RSPO mediated RNF43/ZNRF3 inhibition decreases LRP6 ubiquitination, adding additional exogenous RSPO should rescue the viability. Can you show the outcome of this experiment.

The APCmin culture results represented in Figure S3 are confusing as the conclusion in Figure 3 is that the USP46 complex potentiates Wnt signaling upstream of the destruction complex. Consequently, we would have assumed that USP46 knockdown should not affect the viability of APCmin/- organoids. It seems that the APCmin organoids were derived from freshly isolated tissue. This culture would represent a mixed culture of organoids derived from APCmin/het and APCmin/- cells. The authors should repeat this experiment using sub-cultured, clonal APCmin/- organoids to confirm that USP46 knockdown indeed negatively affects the viability of such organoids.

Reviewer #2 (Remarks to the Author):

Wnt signaling is sensitive to the level of WNT receptors on the cell surface. Identification and elucidation of molecular mechanisms that control WNT receptor turnover is important. In this study, authors discover that the USP46 complex (USP46/UAF1/WDR20) promotes WNT signaling through de-ubiquitinating and stabilizing WNT coreceptor LRP6. Authors show that the USP46 complex regulates WNT signaling in cultured cells, Xenopus embryos and zebrafish embryos. WNT stimulation recruits the USP46 complex to LRP6. Overexpression of the USP46 complex increases the cell surface level of LRP6 and sensitizes cells to WNT signaling. Conversely, knockdown components of this complex decreases LRP6 expression and inhibits WNT signaling. This is a nice and important study. Data are generally convincing. The manuscript is well written and easy to read. I only have several small suggestions for authors to consider.

1. It would be nice to have some general introductions around the USP46 complex and its known functions.
2. Fig. 1A. Authors should mention that coexpression of UAF1 and WDR20 stabilizes USP46. Related to this point, does knockdown UAF1 decrease USP46? It should be indicated in figure legends that USP46 labeled with asterisk is USP46 blot with longer exposure.
3. To rule out potential off-target activity of siRNA, it would be helpful to demonstrate knockout of components of the USP46 complex by CRISPR decreases LRP6 expression and inhibits WNT signaling.
4. Authors might want to comment on the phenotype of zebrafish morpholino experiment. How does it stand against a weak LRP6 morpholino?
5. Fig. 2D/2F. Does lysosomal inhibitor block the effect of USP46 siRNA on LRP6? Authors should confirm USP46 siRNA does not decrease the mRNA level of LRP6.

6. Fig. 5E. This figure is hard to understand. USP46 siRNA is supposed to decrease LRP6 protein level (see Fig. 3). However, in Fig. 5E, it appears that USP46 siRNA increased the total level of LRP6 protein, not just broadening of the band. Maybe the signal can be quantified through fluorescent western blotting. Is this finding related to overexpression of His-Ub? If yes, the experiment should be performed with or without overexpression of His-Ub. Does lysosomal inhibitor affect appearance of LRP6 band with overexpression of His-Ub?

7. Fig. 5F. It would be interesting test whether USP46 siRNA affects viability of intestinal organoids cultured with GSK3 inhibitor.

Reviewer #3 (Remarks to the Author):

Ng and Spencer et al describe a novel function of USP46 to deubiquitylate LRP6, highlighting USP46 as a regulator of WNT receptor turnover and homeostasis. By examining the impact of knocking down components of the destruction complex and measuring β -catenin stabilization, they conclude USP46 works above the level of the destruction complex and go on to show that USP46 acts on LRP6 at the plasma membrane. They show that USP46 regulates LRP6 levels irrespective of exogenous Wnt ligand and deubiquitylates LRP6 which increases its stabilization. This is an interesting study and regulation of the WNT pathway is important in development, homeostasis and tumorigenesis. Overall, I am convinced by the biochemical data, but some of the functional experiments are less robust, particularly in the organoid model. Concerns are listed below

Comments:

1. In Figure 1C the authors show activation of WNT by overexpression of Tri46 in the absence of WNT ligand, however in Figures 1A and 1B, combined expression of these factors does not seem to have an effect on TOPFlash activation. What is the difference between these experiments?

2. Supp Fig 1A shows WNT target genes expression following USP46 silencing – it would be nice to see more than 2 genes profiled here, particularly as the effect size for these targets is relatively small compared to the TOPFlash assay.

3. Figure 3E shows the transcript levels of USP46 in glioblastoma samples from TCGA. The authors state “USP46 was highly correlated with both WDR20 and UAF1 in glioblastoma.” The Pearson coefficients are 0.27 and 0.34 – these are not highly correlated. They are weak to moderate (at best) correlations, even if statistically significant. More importantly, I’m not how sure this data supports the study, particularly if the transcript levels of the genes are not correlated with protein (we don’t know this). Do the tumors that have high expression of all three genes have higher WNT output?

4. Dysregulation of the WNT signaling pathway in cancer by deubiquitylation is not unknown and there are several inhibitors shown in preclinical studies to repress hyperactivation of the WNT signaling pathway. While inhibitors against USP46 may not exist, the authors could comment on the therapeutic potential of targeting USP46 in cancers in which it is overexpressed in the conclusion. It would be nice to see the authors extrapolate further implications for this finding.

5. It is not entirely clear why the authors chose to use the glioblastoma cell lines for validation outside HEK293 cells. Is the USP46 complex and WNT regulation particularly relevant in glioblastoma? My analysis of the TCGA Glioblastoma multiforme does not show any survival difference between cases with high USP46 mRNA. Other than the data in mouse organoids (see below), it would be worth testing whether LRP6 regulation by USP46 impacts WNT activity in classic WNT-driven cancers (CRC?).

6. Intestinal organoids are arguably the best mammalian system to interrogate the functional consequence of WNT deregulation *ex vivo*, yet the data shown in this system is minimal and confusing. While in other systems, USP46 was critical for enabling Wnt-dependent transcriptional responses (Fig S1A, Fig 1E, Fig 3D), the effects in wildtype (extremely WNT-dependent) organoids are surprisingly subtle, showing only 50% decrease in Cell-titer Glo activity (not viability). If USP46 is critical for WNT ligand responses, this assay should reveal a much bigger effect. There is also an inconsistency between the wildtype organoids and APC^{min} experiment, where shRNA1 is more efficient in low dose RSPO, while shRNA2 is more effective in APC^{min} organoids. Do the authors also see an effect of PORCN inhibitors (which completely block receptor-driven WNT signaling) in APC^{min} organoids?

Minor comments

- In the introduction, on page 4, the authors cite MacDonald and Hem (2012) following the statement “Dysregulation of the pathway leads to human diseases such as cancer”. This is actually a review of Fzd/Lpr6 in Wnt signaling that barely touches on cancer. There are more recent and appropriate references for this: Clevers, H. and R. Nusse, Wnt/ β -catenin signaling and disease. *Cell*, 2012. 149(6): p. 1192-205. OR Bugter, J.M., Fenderico, N. & Maurice, M.M. Mutations and mechanisms of WNT pathway tumour suppressors in cancer. *Nat Rev Cancer* 21, 5–21 (2021).

- The authors should better define “hydrodynamic studies”, which is raised in the summary and introduction without any qualification. It is unclear what it means in this context.

- The phrase on p.9 “Having demonstrated that the USP46 complex is required for Wnt signaling in vertebrates...” should probably be more specific to the data that was shown (e.g. Xenopus, zebrafish, and cultured cells)
- Figure 3G and 3H referencing is not correct in the text
- Some level of quantitation for the A172 and U87 data in Figure 3F is important. Is WNT signaling affected?
- ‘The enzymatic activity of USP46 is required for its Wnt activity’ I think this is a typo and should be corrected to ‘The enzymatic activity of USP46 is required for Wnt activity’....
- LRP6 has also been identified to be deubiquitylated by USP19 and was shown to control the stability of many cytoplasmic proteins (PMID: 27751231). The authors should cite this paper and place the study into context as the deubiquitylases that oppose the action of RNF43/ZNF3 are not completely unclear as the authors suggest Giebel et al recently published a model for the regulation of ZNRF3/RNF43 by USP42. While this may be the first study to find a role for USP46 in deubiquitylation, there are other studies which have identified deubiquitylases that target LRP6.

Response to reviewer comments

Reviewer #1 (Remarks to the Author):

This manuscript brings forth a new and important protein complex that positively enhances WNT signalling at the cell membrane. It is mechanistically compelling with in vivo and in vitro relevance. It lacks analyses at the cellular level as is suggested in our comments below:

Analyzing which WNT-driven human organs express the USP46 complex would help to clarify if this WNT control mechanism is global or specific to individual tissues. What are the spatial-temporal expression of each components in developing zebrafish embryos since they are shown to be required for A/P axis formation? Please document their developmental profile.

While we appreciate that detailed studies of the role of the individual components of the USP46 complex in zebrafish development would be very interesting and exciting, we feel that they would be beyond the scope of the current paper, which also provides detailed mechanistic insight into the role of the USP46 complex in regulating Wnt receptor LRP6 activity. In an accompanying paper, we go into detail describing the role of USP46 in *Drosophila*.

While we understand that making triple KO in fish (or *Xenopus*) would be difficult, the use of Morpholinos in these model animals is no longer advised (there are too many off targets effects even when shown to be rescuable). The work by C. Niehrs and colleagues on this important technical shift, also published in *Nature Comm* earlier (<https://www.nature.com/articles/s41467-020-19373-w>), should encourage the authors to revisit some experiments with the use of F0 crispants in fish.

As suggested by Reviewer #1, we have now added USP46 CRISPRi studies in zebrafish to the manuscript, and the results confirmed our MO studies demonstrating both morphological and transcriptional changes consistent with Wnt inhibition.

The study is mostly based on IP and western blot analyses. Showing the subcellular localization and co-localization of the individual USP46 complex components by IF and other means would strengthen the results considerably. Likewise making CRISPR/Cas9 ko cells in 293T cells is now very attainable. The paper relies too much on siRNAs.

We have now added CRISPR/Cas9 experiments to the revised manuscript, as suggested by Reviewer #1. These results are consistent with our finding with siRNA knockdowns.

Figure specific comments:

It seems that the information in figure 1C is entirely duplicated in 1D. In consequence, the authors should remove 1C. In the bar diagram of 1C/D, the value of Wnt3a treated cells in the absence of Tri46 has no error bars. Does this value represent a single read?

If so, I would ask to repeat this experiment and include the average of 3 values for the sake of accuracy.

We apologize for the confusion. All of our TOPFlash Wnt reporter experiments (individually performed in triplicates) were repeated three times or greater. The Wnt3a column contains no error bars because all conditions were graphed relative to the Wnt3a treatment (100% activation). Reviewer #1 is correct in that Fig. 1C and 1D are identical except for the Wnt3a plus Tri46 complex overexpression condition. This latter condition gave such a dramatic increase in TOPFlash reporter activity we believed that it would be difficult for the reader to appreciate the increased TOPFlash reporter response to the expression of the Tri46 complex in the absence of Wnt3a. In retrospect, we agree with Reviewer #1 and have now removed Fig. 1C from the revised manuscript.

In 1E/F, the authors show the effect of USP46 and UAF1 knockdown in human cells. The data of their attempt to knockdown WDR20 is not shown as the knockdown was not efficient. Does the result at least trend in the right direction? Alternatively, are potentially other USP46 co-factors involved in human cells?

There are multiple transcripts of WDR20, and we had difficulty knocking down the expression of WDR20; we tested multiple siRNA and shRNAs but could not detect any decrease in protein levels of WDR20 by immunoblotting. We are happy to report in the revised manuscript that we have successfully performed CRISPR-Cas9 editing of all the USP46 complex components, and we were able to demonstrate decreased β -catenin and LRP6 levels (new Fig. S1C). These results are consistent with our new zebrafish data using CRISPRi to knock down WDR20, which resulted in a Wnt phenotype. Finally, we show in a companion paper that CRISPR-Cas9 knockout of each *Drosophila* USP46 complex component singly also leads to inhibition of Wnt/Wg signaling.

Figure 2a is misleading, there are almost no secondary axes in Wdr20, or Uaf1 or Usp46 (6 out of 101) single injected-embryos compared to control. Why are such low numbers illustrated with duplicated axis images on the right.

Is it statistically significant compared to control (0 out of 159) ? Is the triple o/e assay without effect in a B-cat Morphants? What type of dose curve was tested/established to compare single mRNA vs triple mRNA injections for panels A and B.

Numbers of injected zf embryos in panel C are really too low.

The formation of secondary axes in *Xenopus* embryos is extremely rare (we have never observed it in control embryos). Thus, even low numbers of duplicated axes are significant. Our main goal was to highlight the significant difference in potency between single injections and injecting all three components. We showed representative pictures of duplicated embryos resulting from injections of each USP46 component because we wanted to demonstrate that they did not give rise to unanticipated developmental defects. Using the Fisher exact test, the p-value for WT (0/159) vs. Usp46 (6/101) is $p=0.0031$. For the least affected group, the p-value for Wdr20 (3/84) vs. WT is $p=0.0403$. The number of injected *Xenopus* embryos is typical for classic *Xenopus* axis duplication assays. However, as suggested by Reviewer #1, we now have increased

the number of injected *Xenopus* embryos, and the new p-value is $p=0.0005$ for Wdr20 (8/115) vs. WT (0/257). We have now added this new data to the revised manuscript. Our experiments in cultured human cells indicate that the Usp46 complex acts upstream of the β -catenin destruction complex. Reviewer #1 asked whether the Tri46 complex will affect (or rescue) β -catenin morphants. In addition to the technical difficulty of manipulating the expression of 4 genes simultaneously in *Xenopus* embryos, interpreting the effects of downregulating β -catenin by morpholino injection in the *Xenopus* duplication assay would be complicated. There are several reasons for this: 1) β -catenin mRNA expression does not occur until the mid-blastula transition (after the establishment of the body axis). 2) In *Xenopus* embryos, maternally deposited β -catenin is relegated to the dorsal side of the embryo (MBT is ~6.5–7.5 hr post-fertilization at 20°C) (Collart et al., *Dev Cell* 2017). Because axis duplication assays are performed by injecting the ventral blastomeres, it would be challenging to explore whether the Usp46 complex acts upstream of β -catenin using the axis duplication assay. 3) Because β -catenin is also involved in cell-cell adhesion, it is unlikely that embryos will survive long enough (or exhibit other severe developmental defects) for us to confidently determine whether axis duplication has occurred. We also considered performing the suggested β -catenin morpholino experiment in zebrafish. However, zebrafish express two paralogs of β -catenin in nearly all tissues, further complicating experimentation and interpretation (Zhang et al., *Development* 2012).

For determining the optimal injected amounts of mRNA complex for the axis duplication assay, we initially performed dose-response curves for each component by injecting mRNAs ranging from 0.1 ng to 1 ng. We observed axis duplications at doses as low as 0.5 ng. However, we found that 1 ng was reliably repeatable when injecting the individual components. We then tested injecting the USP46 complex at multiple ratios and concentrations. We found that injecting mRNAs of all three components simultaneously at a 1:1:1 ratio and 1 ng (total) reliably resulted in duplicated axes ($n=3$) and mentioned this in the revised methods section. We agree that the number of injected zebrafish embryos is on the low side, and we have now repeated the experiments with a greater number of injected embryos. These data have now been added to the revised manuscript.

In Figure 5C/D, the authors show that the expression of RNF43 in HEK293 and LF203 cells increased ubiquitylation of LRP6 and that the USP46 complex blocked this effect. ZNRF3 represents an additional E3 ligase for WNT receptors. Does USP46 equally counteract ZNRF3 induced ubiquitination of LRP6?

We agree with Reviewer #1 that it would be interesting to test ZNRF3 since it exhibits similar activity to RNF43 in ubiquitylating Wnt receptors. In the revised manuscript, we now show that overexpression of ZNRF3 increases ubiquitylation of LRP6 and is inhibited upon coexpression of the USP46 complex (new Fig. S5D). Thus, similarly to its effect on RNF43, USP46 counteracts the activity of ZNRF3. We now mention this in our revised Summary.

Figure 5F represents data analyzing the effect of the USP46 complex in intestinal organoids. In general, we would like to understand better which intestinal cells express

all three components of the USP46 complex. Given the WNT gradient along the crypt-villus axis, we would assume the highest USP46 complex levels at the crypt base.

We agree with Reviewer #1 that, given the role of Wnt signaling in normal intestinal growth, it would be interesting to analyze the expression of the USP46 complex components in the intestine. We now show that the USP46 complex is expressed in the human colon and small intestine (new Fig. S7). Immunostaining reveals that the level of membranous USP46 increases towards the crypt base. For UAF1, we found that membranous UAF1 is present in the crypt epithelium and exhibits a similar pattern as USP46. In contrast, we found that WDR20 is uniformly distributed.

The authors show that knockdown of USP46 decreased the viability of intestinal organoids (Fig. 5F). Does knocking down UAF1 or WRD20 has a similar effect on organoid viability? Given that RSPO mediated RNF43/ZNRF3 inhibition decreases LRP6 ubiquitination, adding additional exogenous RSPO should rescue the viability. Can you show the outcome of this experiment.

RSPO is a required component in our media for maintaining intestinal organoids. As Reviewer #1 pointed out, RSPO-mediated RNF43/ZNRF3 inhibition decreases LRP6 ubiquitination and turnover. Thus, increasing the amount of RSPO should counteract the knockdown of USP46. We now show that the knockdown of USP46 dramatically decreases the viability of intestinal organoids when cultured with 2% RSPO conditioned media (CM), which can be inhibited in a dose-dependent manner with increasing amounts of RSP CM (new Fig. S6A, D-F). We observed a similar effect with the knockdown of UAF1. These results have now been added to the revised manuscript.

The APC^{min} culture results represented in Figure S3 are confusing as the conclusion in Figure 3 is that the USP46 complex potentiates Wnt signaling upstream of the destruction complex. Consequently, we would have assumed that USP46 knockdown should not affect the viability of APC^{min}/- organoids. It seems that the APC^{min} organoids were derived from freshly isolated tissue. This culture would represent a mixed culture of organoids derived from APC^{min}/het and APC^{min}/- cells. The authors should repeat this experiment using sub-cultured, clonal APC^{min}/- organoids to confirm that USP46 knockdown indeed negatively affects the viability of such organoids.

We previously showed that loss of LRP6 via shRNA-mediated knockdown resulted in a decreased APC^{min} organoid viability (Saito-Diaz et al., *Dev Cell* 2018). We proposed a model in which LRP6 is activated in APC mutant cells and is required to maintain cell viability. We agree with Reviewer #1 that it could be confusing to the general reader and distract from the main thrust of our story. Thus, we have removed the figure from the revised manuscript.

Reviewer #2 (Remarks to the Author):

Wnt signaling is sensitive to the level of WNT receptors on the cell surface. Identification and elucidation of molecular mechanisms that control WNT receptor turnover is important. In this study, authors discover that the USP46 complex (USP46/UAF1/WDR20) promotes WNT signaling through de-ubiquitinating and

stabilizing WNT coreceptor LRP6. Authors show that the USP46 complex regulates WNT signaling in cultured cells, *Xenopus* embryos and zebrafish embryos. WNT stimulation recruits the USP46 complex to LRP6. Overexpression of the USP46 complex increases the cell surface level of LRP6 and sensitizes cells to WNT signaling. Conversely, knockdown components of this complex decreases LRP6 expression and inhibits WNT signaling. This is a nice and important study. Data are generally convincing. The manuscript is well written and easy to read. I only have several small suggestions for authors to consider.

1. It would be nice to have some general introductions around the USP46 complex and its known functions.

We thank Reviewer #2 for the suggestion. We have now added more detail about the USP46 complex and its known functions in the revised manuscript.

2. Fig. 1A. Authors should mention that coexpression of UAF1 and WDR20 stabilizes USP46. Related to this point, does knockdown UAF1 decrease USP46? It should be indicated in figure legends that USP46 labeled with asterisk is USP46 blot with longer exposure.

We agree with Reviewer #2's suggestion, and we now comment on the stabilizing effects of expressing UAF1 and WDR20 on USP46 in the revised manuscript. It was previously shown that UAF1 promotes the stability of USP46 (Hodul et al., *J Bio Chem* 2020), and we now show this in the revised manuscript (Fig. S1B). In addition, we indicated that the immunoblot of USP46 labeled with an asterisk is a long exposure in the figure legend of the revised manuscript.

3. To rule out potential off-target activity of siRNA, it would be helpful to demonstrate knockout of components of the USP46 complex by CRISPR decreases LRP6 expression and inhibits WNT signaling.

We agree with Reviewer #2 on this point, and we have now performed CRISPR-Cas9 editing of all the components in the complex (new Fig. S1C). To minimize compensation that may occur with the selection of single clones, we performed knockout in a population of cells and followed levels of b-catenin and LRP6 over time. We now show that knockout of USP46, UAF1, and WDR20 all resulted in reduced b-catenin within six days of treatment with their gRNA. These results have now been added to the revised manuscript.

4. Authors might want to comment on the phenotype of zebrafish morpholino experiment. How does it stand against a weak LRP6 morpholino?

We are reluctant to comment on the LRP6 morphant phenotype in zebrafish because the eye morphology has not been described in detail in published reports. There are three papers in *Zfin* that describe the MO-mediated knockdown of LRP6 in zebrafish. Two of these papers (Jiang et al., *The EMBO Journal* 2012 and Shi et al., *Birth Defects Research* 2017) show photos of whole zebrafish and stated observation of dorsoventralized phenotypes using MO at 3 ng. Based on the morphant photos presented, we estimate that they have class 3 to class 5 cyclopia on the scale by Marlow et al., (*Dev Biol* 1998).

5. Fig. 2D/2F. Does lysosomal inhibitor block the effect of USP46 siRNA on LRP6? Authors should confirm USP46 siRNA does not decrease the mRNA level of LRP6. We have now performed the experiments suggested by Reviewer #2. We show that the macrolide antibiotic, bafilomycin A, blocks the effect of USP46 siRNA on LRP6 turnover, suggesting a lysosomal mechanism (new Fig. S5F). This result is consistent with the degradation of ubiquitinated LRP6 receptors via the lysosomal pathway (Perrody et al., *eLife*, 2016). In addition, we also show that knocking down USP46 by siRNA does not decrease LRP6 mRNA levels (new Fig. S1A).

6. Fig. 5E. This figure is hard to understand. USP46 siRNA is supposed to decrease LRP6 protein level (see Fig. 3). However, in Fig. 5E, it appears that USP46 siRNA increased the total level of LRP6 protein, not just broadening of the band. Maybe the signal can be quantified through fluorescent western blotting. Is this finding related to overexpression of His-Ub? If yes, the experiment should be performed with or without overexpression of His-Ub. Does lysosomal inhibitor affect appearance of LRP6 band with overexpression of His-Ub?

Unfortunately, we could not perform fluorescent western blotting for technical reasons. As suggested by Reviewer #2, we tested overexpression of His₆-Ub versus a control plasmid (pcDNA) and now show that detection of LRP6 ubiquitylation (enhanced with USP46 siRNA treatment) in the His₆-Ub assay is specific for cells that were transfected with His₆-Ub (New Fig. S5E). We found that the broadened protein bands on immunoblots are challenging to quantify. In this new experiment (added to the revised manuscript), the signal for LRP6 did not exhibit broadening, and the steady-state level of LRP6 in the lysate was observably reduced. Finally, we found that bafilomycin A treatment stabilized LRP6 in basal and USP46 siRNA-treated cells that overexpressed His₆-Ub (new Fig. S5F).

7. Fig. 5F. It would be interesting test whether USP46 siRNA affects viability of intestinal organoids cultured with GSK3 inhibitor.

We agree with Reviewer #2 that this is an interesting experiment to test. We found that when intestinal organoids are cultured with the GSK3 inhibitor, CHIR99021, shRNA knockdown of USP46 resulted in a statistically significant decrease in the viability of the intestinal organoids (albeit it was not very dramatic). CHIR99021 blocks b-catenin phosphorylation (and subsequent degradation) and mimics the activated b-catenin mutant state. A previous study from the Boutros lab demonstrated that signaling from Wnt receptors was necessary to maintain supraphysiological Wnt signaling in colorectal mutant cancer cells for viability, even with b-catenin mutation (Voloshanenکو et al., *Nat Commun* 2013). Thus, it is possible that the knockdown of USP46 would reduce the viability of CHIR99021 treated cells by decreasing LRP6 levels. Because this result could be confusing to the general reader and distract from the main focus of our story, we have opted to not include this result in the revised manuscript.

Reviewer #3 (Remarks to the Author):

Ng and Spencer et al describe a novel function of USP46 to deubiquitylate LRP6,

highlighting USP46 as a regulator of WNT receptor turnover and homeostasis. By examining the impact of knocking down components of the destruction complex and measuring b-catenin stabilization, they conclude USP46 works above the level of the destruction complex and go on to show that USP46 acts on LRP6 at the plasma membrane. They show that USP46 regulates LRP6 levels irrespective of exogenous Wnt ligand and deubiquitylates LRP6 which increases its stabilization. This is an interesting study and regulation of the WNT pathway is important in development, homeostasis and tumorigenesis. Overall, I am convinced by the biochemical data, but some of the functional experiments are less robust, particularly in the organoid model. Concerns are listed below

Comments:

1. In Figure 1C the authors show activation of WNT by overexpression of Tri46 in the absence of WNT ligand, however in Figures 1A and 1B, combined expression of these factors does not seem to have an effect on TOPFlash activation. What is the difference between these experiments?

We apologize for the confusion. The effects of overexpressing Tri46 in the presence of Wnt3a were so dramatic that it obscured the effect of Tri46 on Wnt reporter activity in the absence of Wnt3a. To determine whether overexpressing Tri46 was sufficient to activate Wnt reporter activity in HEK393 cells, we repeated the experiment in the absence of Wnt and compared activation to Wnt3a treated cells (Fig. 1C and D). Because Fig. 1C and 1D are essentially the same experiments, we have now removed Fig. 1C in the revised manuscript. For Fig. 1A, we tested overexpression of each of the three individual components, whereas, for Fig. 1B, we tested pair-wise combinations of the Tri46 complex. We now clarify this point in the revised manuscript.

2. Supp Fig 1A shows WNT target genes expression following USP46 silencing – it would be nice to see more than 2 genes profiled here, particularly as the effect size for these targets is relatively small compared to the TOPFlash assay.

As suggested by Reviewer #3, we have added the expression of another Wnt target gene, *Dkk1* (new Fig. S1A). We now show that downregulating USP46 results in decreased *Dkk1* expression when compared to the non-targeting control.

3. Figure 3E shows the transcript levels of USP46 in glioblastoma samples from TCGA. The authors state “USP46 was highly correlated with both WDR20 and UAF1 in glioblastoma.” The Pearson coefficients are 0.27 and 0.34 – these are not highly correlated. They are weak to moderate (at best) correlations, even if statistically significant. More importantly, I’m not how sure this data supports the study, particularly if the transcript levels of the genes are not correlated with protein (we don’t know this). Do the tumors that have high expression of all three genes have higher WNT output?

We agree with Reviewer #3 that the correlations amongst USP46, WDR20, and UAF1 are moderate and now state this in the text. In the revised manuscript, we examined the expression of the Wnt target genes, *Nkd1* and *Axin2*, in TCGA-GBM tumors that

express high mRNA levels of USP46, UAF1, and WDR20 versus TCGA-GBM tumors that express low mRNA levels of USP46, UAF1, and WDR20. We found that TCGA-GBM tumors with high USP46, UAF1, and WDR20 mRNA levels had significantly higher *Nkd1* and *Axin2* expression when compared to TCGA-GBM tumors with low USP46, UAF1, and WDR20 mRNA levels. We have now added this new data to the revised manuscript (new Fig. S9).

4. Dysregulation of the WNT signaling pathway in cancer by deubiquitylation is not unknown and there are several inhibitors shown in preclinical studies to repress hyperactivation of the WNT signaling pathway. While inhibitors against USP46 may not exist, the authors could comment on the therapeutic potential of targeting USP46 in cancers which it is overexpressed in the conclusion. It would be nice to see the authors extrapolate further implications for this finding.

We thank Reviewer #3 for the suggestion. In the revised manuscript, we now comment on the potential of targeting the USP46 complex in cancers where its overexpression (or elevated levels of Wnt signaling due to receptor activation) may play a role in tumorigenesis.

5. It is not entirely clear why the authors chose to use the glioblastoma cell lines for validation outside HEK293 cells. Is the USP46 complex and WNT regulation particularly relevant in glioblastoma? My analysis of the TCGA Glioblastoma multiforme does not show any survival difference between cases with high USP46 mRNA. Other than the data in mouse organoids (see below), it would be worth testing whether LRP6 regulation by USP46 impacts WNT activity in classic WNT-driven cancers (CRC?).

Because USP46 is highly expressed in the brain, we choose to explore the effect of knocking down USP46 in glioblastoma lines as we expect that its levels will also be elevated. We have now explored the loss of USP46 in a CRC line (DLD1), which has a truncating mutation in APC, and show that downregulating USP46 in this CRC line also results in decreased LRP6 levels (Fig. S3C), which would be expected to impact Wnt-receptor mediated signaling in the CRC line, based on the detailed analysis of Wnt ligand signaling in CRCs from the Boutros lab (Voloshanenko et al., *Nat Commun* 2013).

6. Intestinal organoids are arguably the best mammalian system to interrogate the functional consequence of WNT deregulation *ex vivo*, yet the data shown in this system is minimal and confusing. While in other systems, USP46 was critical for enabling Wnt-dependent transcriptional responses (Fig S1A, Fig 1E, Fig 3D), the effects in wildtype (extremely WNT-dependent) organoids are surprisingly subtle, showing only 50% decrease in Cell-titer Glo activity (not viability). If USP46 is critical for WNT ligand responses, this assay should reveal a much bigger effect. There is also an inconsistency between the wildtype organoids and APC^{min} experiment, where shRNA1 is more efficient in low dose RSPO, while shRNA2 is more effective in APC^{min} organoids. Do the authors also see an effect of PORCN inhibitors (which completely block receptor-driven WNT signaling) in APC^{min} organoids?

Our previous study (Saito-Diaz et al., *Dev Cell* 2018) showed that Wnt receptors were activated in APC mutant cells. Furthermore, we rigorously demonstrated that adding porcupine inhibitors (or knocking out porcupine) did not inhibit Wnt receptor activation in APC mutant CRC cells. Thus, receptor activation in APC mutant cells is ligand-independent. Our recent study indicates that the porcupine inhibitor, LGK-974, does not affect the growth of *APC^{min}* tumors, consistent with our earlier results. In our revised manuscript, we decided not to add these data and to remove our other *APC^{min}* organoid study because we believe it would make the story more confusing to the general reader and distract from the main thrust of our story.

Minor comments

- In the introduction, on page 4, the authors cite MacDonald and Hem (2012) following the statement “Dysregulation of the pathway leads to human diseases such as cancer”. This is actually a review of Fzd/Lpr6 in Wnt signaling that barely touches on cancer. There are more recent and appropriate references for this: Clevers, H. and R. Nusse, Wnt/ β -catenin signaling and disease. *Cell*, 2012. 149(6): p. 1192-205. OR Bugter, J.M., Fenderico, N. & Maurice, M.M. Mutations and mechanisms of WNT pathway tumour suppressors in cancer. *Nat Rev Cancer* 21, 5–21 (2021).

We agree that the Clevers and Nusse review is more appropriate and have made the suggested change by Reviewer #3.

- The authors should better define “hydrodynamic studies”, which is raised in the summary and introduction without any qualification. It is unclear what it means in this context.

Hydrodynamic studies classically refer to gel filtration and/or sucrose gradient centrifugation. To clarify, we have now specified size-exclusion chromatography and sucrose density gradient centrifugation in the revised manuscript.

- The phrase on p.9 “Having demonstrated that the USP46 complex is required for Wnt signaling in vertebrates...” should probably be more specific to the data that was shown (e.g. *Xenopus*, zebrafish, and cultured cells)

We have rewritten the text to be more specific, as suggested by the reviewer.

- Figure 3G and 3H referencing is not correct in the text

We thank Reviewer #3 for the oversight. We have now made the correction.

- Some level of quantitation for the A172 and U87 data in Figure 3F is important. Is WNT signaling affected?

We have moved this figure to the supplement section. We have quantified the changes in LRP6 levels and demonstrated that they are significantly reduced when USP46 is knocked down. Unfortunately, we had difficulties assessing the level of Wnt signaling in these two cell lines using our Wnt reporter assays. We believe that it is more significant that triple-high USP46/WDR48/WDR20 glioblastoma tumors exhibit significantly higher

expression of Wnt target genes compared to triple-low USP46/WDR48/WDR20 glioblastoma tumors.

- ‘The enzymatic activity of USP46 is required for its Wnt activity’ I think this is a typo and should be corrected to ‘The enzymatic activity of USP46 is required for Wnt activity’....

We have now made the changes suggested by Reviewer #3 in the revised text.

- LRP6 has also been identified to be deubiquitylated by USP19 and was shown to control the stability of many cytoplasmic proteins (PMID: 27751231). The authors should cite this paper and place the study into context as the deubiquitylases that oppose the action of RNF43/ZNF3 are not completely unclear as the authors suggest Giebel et al recently published a model for the regulation of ZNRF3/RNF43 by USP42. While this may be the first study to find a role for USP46 in deubiquitylation, there are other studies which have identified deubiquitylases that target LRP6.

Our studies with USP46 shows that it operates at the plasma membrane to regulate LRP6 turnover. We failed to mention USP19 because it appears that USP19 regulates the maturation of LRP6 in the ER. We also did not mention USP42 because it acts on the E3 ligases, RNF43/ZNRF3. For completion, we have now added statements about the roles of UPS19 and USP42 in the revised manuscript as suggested by Reviewer #3.

REVIEWER COMMENTS

Reviewer #1 (Remarks to the Author):

Besides the 3 comments below, the authors have done an earnest revision addressing most of our comments.

We strongly suggested a developmental QPCR for the USP46 components during zebrafish embryogenesis. This is not a difficult task and will be needed to support the specificity of the CRISPRi or MO-mediated knockdown of these genes leading to cyclopia. Please provide in wt, and CRISPRi-injected embryos in FigS2.

We have not been shown data which is meant to support this work as is stated by authors in the rebuttal:

'Finally, we show in a companion paper that CRISPR-Cas9 knockout of each Drosophila USP46 complex component singly also leads to inhibition of Wnt/Wg signaling.'

We cannot comment, let alone vouch for this, solely based on the authors' claims that this exists somewhere else.

No efforts, despite our encouragements, were made to visualize by IF in cultured cells (eg 293T cells) LRP6 with the studied USP46 components. Why is that ?

Reviewer #2 (Remarks to the Author):

Authors have sufficiently addressed my concerns.

Reviewer #3 (Remarks to the Author):

The authors have reasonably addressed my concerns raised in the last round of review. No further concerns

We would like to thank the reviewers for their time and effort. Both Reviewers #2 and #3 were satisfied with our response and had no further concerns. Review #1 commented on our attempt to address most of the issues raised but requested that we address the specificity of our knockdown studies, which we have now done.

REVIEWER COMMENTS

Reviewer #1 (Remarks to the Author):

Besides the 3 comments below, the authors have done an earnest revision addressing most of our comments.

1. We strongly suggested a developmental QPCR for the USP46 components during zebrafish embryogenesis. This is not a difficult task and will be needed to support the specificity of the CRISPRi or MO-mediated knockdown of these genes leading to cyclopia. Please provide in wt, and CRISPRi-injected embryos in FigS2.

We agree that this is an important experiment to show specificity. In the revised manuscript, we have now demonstrated (using two different primer sets) that CRISPRi-injected embryos show a significant reduction in the mRNAs of the targeted USP46 complex component (new Figure S3) compared to injected controls (wild-type and Cas9 only).

2. We have not been shown data which is meant to support this work as is stated by authors in the rebuttal:

'Finally, we show in a companion paper that CRISPR-Cas9 knockout of each Drosophila USP46 complex component singly also leads to inhibition of Wnt/Wg signaling.'

We cannot comment, let alone vouch for this, solely based on the authors' claims that this exists somewhere else.

2. No efforts, despite our encouragements, were made to visualize by IF in cultured cells (eg 293T cells) LRP6 with the studied USP46 components. Why is that ?

We have attempted to perform the experiment suggested by Reviewer #1, but we were unsuccessful in observing clear co-localization between LRP6 and the USP46 complex in the presence or absence of Wnt3a. This was due to the fact that commercial antibodies did not work very well in our hands to stain endogenous LRP6 and that a large amount of UPS46 complexes are involved in other cellular pathways (and remained cytoplasmic), likely overwhelming the signal for the portion that is bound to LRP6. Regardless, we believe that our biochemical studies showed a clear association between the USP46 complex and LRP6, providing strong evidence for their interaction.

Reviewer #2 (Remarks to the Author):

Authors have sufficiently addressed my concerns.

Reviewer #3 (Remarks to the Author):

The authors have reasonably addressed my concerns raised in the last round of review. No further concerns